# Hybrid Variance-Reduced SGD Algorithms For Minimax Problems with Nonconvex-Linear Function

**Quoc Tran-Dinh**[*]   **Deyi Liu**[*]   **Lam M. Nguyen**[†]

[*]Department of Statistics and Operations Research
The University of North Carolina at Chapel Hill, Chapel Hill, NC 27599
Emails: {quoctd@email.unc.edu, deyi.liu@live.unc.edu}

[†]IBM Research, Thomas J. Watson Research Center
Yorktown Heights, NY10598, USA.
Email: lamnguyen.mltd@ibm.com

## Abstract

We develop a novel and single-loop variance-reduced algorithm to solve a class of stochastic nonconvex-convex minimax problems involving a nonconvex-linear objective function, which has various applications in different fields such as machine learning and robust optimization. This problem class has several computational challenges due to its nonsmoothness, nonconvexity, nonlinearity, and non-separability of the objective functions. Our approach relies on a new combination of recent ideas, including smoothing and hybrid biased variance-reduced techniques. Our algorithm and its variants can achieve $\mathcal{O}(T^{-2/3})$-convergence rate and the best known oracle complexity under standard assumptions, where $T$ is the iteration counter. They have several computational advantages compared to existing methods such as simple to implement and less parameter tuning requirements. They can also work with both single sample or mini-batch on derivative estimators, and with constant or diminishing step-sizes. We demonstrate the benefits of our algorithms over existing methods through two numerical examples, including a nonsmooth and nonconvex-non-strongly concave minimax model.

## 1 Introduction

We study the following stochastic minimax problem with nonconvex-linear objective function, which covers various practical problems in different fields, see, e.g., [4, 10, 12]:

$$\min_{x \in \mathbb{R}^p} \max_{y \in \mathbb{R}^n} \left\{ \Psi(x,y) := \mathcal{R}(x) + \mathbb{E}_\xi \big[ \langle Ky, \mathbf{F}(x,\xi) \rangle \big] - \psi(y) \right\}, \tag{1}$$

where $\mathbf{F} : \mathbb{R}^p \times \Omega \to \mathbb{R}^q$ is a stochastic vector function defined on a probability space $(\Omega, \mathbb{P})$, $K \in \mathbb{R}^{q \times n}$ is a given matrix, $\langle \cdot, \cdot \rangle$ is an inner product, and $\psi : \mathbb{R}^n \to \mathbb{R} \cup \{+\infty\}$ and $\mathcal{R} : \mathbb{R}^p \to \mathbb{R} \cup \{+\infty\}$ are proper, closed, and convex functions [3]. Problem (1) is a special case of the nonconvex-concave minimax problem, where $\mathcal{H}(x,y) := \mathbb{E}_\xi \big[ \langle Ky, \mathbf{F}(x,\xi) \rangle \big]$ is nonconvex in $x$ and linear in $y$.

Due to the linearity of $\mathcal{H}$ w.r.t. $y$, (1) can be reformulated into a general stochastic compositional nonconvex problem of the form:

$$\min_{x \in \mathbb{R}^p} \left\{ \Psi_0(x) := \phi_0(F(x)) + \mathcal{R}(x) \equiv \phi_0 \big( \mathbb{E}_\xi [\mathbf{F}(x,\xi)] \big) + \mathcal{R}(x) \right\}, \tag{2}$$

where $\phi_0$ is a convex, but possibly nonsmooth function, defined as

$$\phi_0(u) := \max_{y \in \mathbb{R}^n} \left\{ \langle K^\top u, y \rangle - \psi(y) \right\} \equiv \psi^*(K^\top u), \tag{3}$$

with $\psi^*$ being the Fenchel conjugate of $\psi$ [3], and we define $\Phi_0(x) := \phi_0(F(x))$. Note that problem (2) is completely different from existing models such as [8, 9], where the expectation is inside the outer function $\phi_0$, i.e., $\phi_0\left(\mathbb{E}_\xi\left[\mathbf{F}(x, \xi)\right]\right)$. We refer to this setting as a "non-separable" model.

**Challenges:** Developing numerical methods for solving (1) or (2) faces several challenges. First, it is often nonconvex, i.e., $F$ is not affine. Many recent papers consider special cases of (2) when $\Psi_0$ in (2) is convex by imposing restrictive conditions, which are unfortunately not realistic in applications. Second, the max-form $\phi_0$ in (3) is often nonsmooth if $\psi$ is not strongly convex. This prevents the use of gradient-based methods. Third, since the expectation is inside $\phi_0$, it is very challenging to form an unbiased estimate for [sub]gradients of $\Phi_0$, making classical stochastic gradient-based methods inapplicable. Finally, prox-linear operator-based methods as in [8, 9, 34, 45] require large mini-batch evaluations of both function value $\mathbf{F}$ and its Jacobian $\mathbf{F}'$, see [34, 43, 45], instead of single sample or small mini-batch, making them less flexible and more expensive than gradient-based methods.

**Related work:** Problem (1) has recently attracted considerable attention due to key applications, e.g., in game theory, robust optimization, distributionally robust optimization, and generative adversarial nets (GANs) [4, 10, 12, 30]. Various first-order methods have been developed to solve (1) during the past decades for both convex-concave models , e.g., [3, 14, 23, 36] and nonconvex-concave settings [17, 18, 20, 28, 32]. Some recent works consider a nonnonvex-nonconcave formulation, e.g., [27, 40]. However, they still rely on additional assumptions to guarantee that the maximization problem in (3) can globally be solved. One well-known assumption is the Polyak-Łojasiewicz (PL) condition, which is rather strong and often used to guarantee linear convergence rates. A majority of these works focus on deterministic models, while some methods have been extended to stochastic settings, e.g., [17, 40]. Although (1) is a special case of a general model in [17, 18, 40], it almost covers all examples in [17, 40]. Compared to these, we only consider a special class of minimax problems where the function $\mathcal{H}$ is linear in $y$. However, our algorithm is rather simple with a single loop, and our oracle complexity is significantly improved over the ones in [17, 40].

In a very recent work [21], which is concurrent to our paper, the authors develop a double-loop algorithm, called SREDA, to handle a more general case than (1) where $\mathcal{H}$ is strongly concave in $y$. Their method exploits the SARAH estimator introduced in [26] and can achieve the same $\mathcal{O}\left(\varepsilon^{-3}\right)$ oracle complexity as ours in Theorem 3.1 below. Compared to our work here, though the problem setting in [21] is more general than (1), it does not cover the non-strongly convex case. This is important to handle stochastic constrained optimization problems, where $\psi$ is nonsmooth and convex, but not necessarily strongly convex (see, e.g., (32) below as an example). Moreover, the SREDA algorithm in [21] requires double loops with large mini-batch sizes in both function values and derivatives and uses small learning rates to achieve the desired oracle complexity.

It is interesting that the minimax problem (1) can be reformulated into a nonconvex compositional optimization problem of the form (2). The formulation (2) has been broadly studied in the literature under both deterministic and stochastic settings, see, e.g., [8, 9, 15, 25, 33, 37]. If $q = 1$ and $\phi_0(u) = u$, then (2) reduces to the standard stochastic optimization model studied e.g., in [11, 29]. In the deterministic setting, one common method to solve (2) is the prox-linear-type method, which is also known as a Gauss-Newton method [15, 25]. This method has been studied in several papers, including [8, 9, 15, 25, 33]. However, the prox-linear operator often does not have a closed form expression, and its evaluation may require solving a general nonsmooth strongly convex subproblem.

In the stochastic setting as (2), [37, 38] proposed stochastic compositional gradient methods to solve more general forms than (2), but they required a set of stronger assumptions than Assumptions 2.1-2.2 below, including the smoothness of $\phi_0$. Recent related works include [16, 19, 39, 41, 42], which also rely on similar ideas. For instance, [17] proposed a double loop subgradient-based method with $\mathcal{O}\left(\varepsilon^{-6}\right)$ oracle complexity. Another subgradient-based method was recently proposed in [40] based on a two-side PL condition. Stochastic methods exploiting prox-linear operators have also been recently proposed in [34, 45], which are essentially extensions of existing deterministic methods to (2). Together with algorithms, convergence guarantees, stochastic oracle complexity bounds have also been estimated. For instance, [37] obtained $\mathcal{O}\left(\varepsilon^{-8}\right)$ oracle complexity for (2), while it was improved to $\mathcal{O}\left(\varepsilon^{-4.5}\right)$ in [38]. Recent works [43, 44] further improved the complexity to $\mathcal{O}\left(\varepsilon^{-3}\right)$. These methods require the smoothness of both $\phi_0$ and $F$, use large batch sizes, and need a double-loop scheme. In contrast, ***our method has single loop, can work with either single sample or mini-batch, and allows both constant or diminishing step-sizes***. For nonsmooth $\phi_0$, under the same assumptions as [34, 45], our methods achieve $\mathcal{O}\left(\varepsilon^{-3}\right)$ Jacobian and $\mathcal{O}\left(\varepsilon^{-5}\right)$ function evaluation complexity as in

those papers. However, our method is gradient-based, which only uses proximal operator of $\psi$ and $\mathcal{R}$ instead of a complex prox-linear operator as in [34, 45]. Note that even if $\psi$ and $\mathcal{R}$ have closed-form proximal operator, the prox-linear operator still does not have a closed-form solution, and requires to solve a composite and possibly nonsmooth strongly convex subproblem involving a linear operator, see, e.g., [34]. Moreover, our method can work with both single sample and mini-batch for Jacobian $\mathbf{F}'$ compared to a large batch size as in [34, 45].

**Our contribution:** Our main contribution in this paper can be summarized as follows:

(a) We develop a new single-loop hybrid variance-reduced SGD algorithm to handle (1) under Assumptions 2.1 and 2.2 below. Under the strong convexity of $\psi$, our algorithm has $\mathcal{O}\left((bT)^{-2/3}\right)$ convergence rate to approximate a KKT (Karush-Kuhn-Tucker) point of (1), where $b$ is the batch size and $T$ is the iteration counter. We also estimate an $\mathcal{O}\left(\varepsilon^{-3}\right)$-oracle complexity to obtain an $\varepsilon$-KKT point, matching the best known one as, e.g., in [21, 43, 44]. Our complexity bound holds for a wide range of $b$ as opposed to a specific choice as in [21, 43, 44]. Moreover, our algorithm has only a single loop compared to [21, 43]..

(b) When $\psi$ is non-strongly convex, we combine our approach with a smoothing technique to develop a gradient-based variant, that can achieve the best-known $\mathcal{O}\left(\varepsilon^{-3}\right)$ Jacobian and $\mathcal{O}\left(\varepsilon^{-5}\right)$ function evaluations of $\mathbf{F}$ for finding an $\varepsilon$-KKT point of (1). Moreover, our algorithm does not require prox-linear operators and large batches for Jacobian as in [34, 45].

(c) We also propose a simple restarting technique without sacrificing convergence guarantees to accelerate the practical performance of both cases (a) and (b) (see Supp. Doc. C).

Our methods exploit a recent biased hybrid estimators introduced in [35] as opposed to SARAH ones in [34, 43, 45]. This allows us to simplify our algorithm with a single loop and without large batches at each iteration compared to [43]. As indicated in [2], our $\mathcal{O}\left(\varepsilon^{-3}\right)$ oracle complexity is optimal under the considered assumptions. If $\psi$ is non-strongly convex (i.e. $\phi_0$ in (2) can be nonsmooth), then our algorithm is fundamentally different from the ones in [34, 45] as it does not use prox-linear operator. Note that evaluating a prox-linear operator requires to solve a general strongly convex but possible nonsmooth subproblem. In addition, they only work with large batch sizes of both $\mathbf{F}$ and $\mathbf{F}'$.

**Content:** Section 2 states our assumptions and recalls some mathematical tools. Section 3 develops a new algorithm and analyzes its convergence. Section 4 provides two numerical examples to compare our methods. All technical details and proofs are deferred to Supplementary Document (Supp. Doc.).

## 2 Basic assumptions, KKT points and smoothing technique

**Notation:** We work with finite-dimensional space $\mathbb{R}^p$ equipped with standard inner product $\langle \cdot, \cdot \rangle$ and Euclidean norm $\| \cdot \|$. For a function $\phi : \mathbb{R}^p \to \mathbb{R} \cup \{+\infty\}$, $\mathrm{dom}(\phi)$ denotes its domain. If $\phi$ is convex, then $\mathrm{prox}_\phi$ denotes its proximal operator, $\partial \phi$ denotes its subdifferential, and $\nabla \phi$ is its [sub]gradient, see, e.g., [3]. $\phi$ is $\mu_\phi$-strongly convex with a strongly convex parameter $\mu_\phi > 0$ if $\phi(\cdot) - \frac{\mu_\phi}{2}\| \cdot \|^2$ remains convex. For a smooth vector function $F : \mathbb{R}^p \to \mathbb{R}^q$, $F'$ denotes its Jacobian. We use $\mathrm{dist}\,(x, \mathcal{X}) := \inf_{y \in \mathcal{X}} \|x - y\|$ to denote the Euclidean distance from $x$ to a convex set $\mathcal{X}$.

### 2.1 Model assumptions

Let $F(x) := \mathbb{E}_\xi\big[\mathbf{F}(x, \xi)\big]$ denote the expectation function of $\mathbf{F}$ and $\mathrm{dom}(\Psi_0)$ denote the domain of $\Psi_0$. Throughout this paper, we always assume that $\Psi_0^\star := \inf_{x \in \mathbb{R}^p}\{\Psi_0(x) := \phi_0(F(x)) + \mathcal{R}(x)\} > -\infty$ in (2) and $\mathcal{R}$ is proper, closed, and convex without recalling them in the sequel. Our goal is to develop stochastic gradient-based algorithms to solve (1) relying on the following assumptions:

**Assumption 2.1.** The function $\mathbf{F}$ in problem (1) or (2) satisfies the following assumptions:

(a) **Smoothness:** $\mathbf{F}(\cdot, \cdot)$ is $L_F$-average smooth with $L_F \in (0, +\infty)$, i.e.:
$$\mathbb{E}_\xi\big[\,\|\mathbf{F}'(x, \xi) - \mathbf{F}'(y, \xi)\|^2\,\big] \leq L_F^2\,\|x - y\|^2, \quad \forall x, y \in \mathrm{dom}(\Psi_0). \tag{4}$$

(b) **Bounded variance:** There exists two constants $\sigma_F, \sigma_J \in (0, +\infty)$ such that
$$\mathbb{E}_\xi\big[\,\|\mathbf{F}(x, \xi) - F(x)\|^2\,\big] \leq \sigma_F^2 \quad \text{and} \quad \mathbb{E}_\xi\big[\,\|\mathbf{F}'(x, \xi) - F'(x)\|^2\,\big] \leq \sigma_J^2, \quad \forall x \in \mathrm{dom}(\Psi_0).$$

(c) **Lipschitz continuity:** $F(\cdot)$ is $M_F$-average Lipschitz continuous with $M_F \in (0, +\infty)$, i.e.:
$$\mathbb{E}_\xi\big[\,\|\mathbf{F}'(x, \xi)\|^2\,\big] \leq M_F^2, \quad \forall x \in \mathrm{dom}(\Psi_0). \tag{5}$$

Note that Assumptions 2.1 are standard in stochastic nonconvex optimization, see [34, 43, 44, 45]. If $\mathrm{dom}(\mathcal{R})$ is bounded, then $\mathrm{dom}(\Psi_0)$ is bounded, and this assumption automatically holds.

For $\psi$, we only require the following assumption, which is mild and holds for many applications.

**Assumption 2.2.** The function $\psi$ in (1) is proper, closed, and convex. Moreover, $\mathrm{dom}(\psi)$ is bounded by $M_\psi \in (0, +\infty)$, i.e.: $\sup\{\|y\| : y \in \mathrm{dom}(\psi)\} \leq M_\psi$.

An important special case of $\psi$ is the indicator of convex and bounded sets. Hitherto, we do not require $\phi_0$ and $\mathcal{R}$ in (2) to be smooth or strongly convex. They can be nonsmooth so that (2) can also cover constrained problems. Note that the boundedness of $\mathrm{dom}(\psi)$ is equivalent to the Lipschitz continuity of $\phi_0$ (Lemma A.1). Simple examples of $\phi_0$ include norms and gauge functions.

## 2.2 KKT points and approximate KKT points

Since (1) is nonconvex-concave, a pair $(x^\star, y^\star)$ is said to be a KKT point of (1) if

$$0 \in F'(x^\star)^\top K y^\star + \partial\mathcal{R}(x^\star) \qquad \text{and} \qquad 0 \in K^\top F(x^\star) - \partial\psi(y^\star). \tag{6}$$

From (6), we have $y^\star \in \partial\psi^*(K^\top F(x^\star))$. Substituting this $y^\star$ into the first expression, we get

$$0 \in F'(x^\star)^\top \partial\phi_0(F(x^\star)) + \partial\mathcal{R}(x^\star). \tag{7}$$

Here, we have used $K^\top \partial\psi^*(K^\top u) = \partial\phi_0(u)$, where $\phi_0$ is given by (3) This inclusion shows that $x^\star$ is a stationary point of (2). In the convex-concave case, under mild assumptions, a KKT point is also a saddle-point of (1). In particular, if (2) is convex, then $x^\star$ is also its global optimum of (2).

However, in practice, we can only find an approximation $(\tilde{x}_0^*, \tilde{y}_0^*)$ of a KKT point $(x^\star, y^\star)$ for (1).

**Definition 2.1.** Given any tolerance $\varepsilon > 0$, $(\tilde{x}_0^*, \tilde{y}_0^*)$ is called an $\varepsilon$-KKT point of (1) if

$$\mathbb{E}\big[\mathcal{E}(\tilde{x}_0^*, \tilde{y}_0^*)\big] \leq \varepsilon,$$

$$\text{where} \quad \mathcal{E}(x, y) \quad := \mathrm{dist}\big(0, F'(x)^\top K y + \partial\mathcal{R}(x)\big) + \mathrm{dist}\big(0, K^\top F(x) - \partial\psi(y)\big). \tag{8}$$

Here, the expectation is taken overall the randomness from both model (1) and the algorithm. Clearly, if $\mathbb{E}\big[\mathcal{E}(\tilde{x}_0^*, \tilde{y}_0^*)\big] = 0$, then $(\tilde{x}_0^*, \tilde{y}_0^*)$ is a KKT point of (1) as characterized by (6).

## 2.3 Smoothing techniques

Under Assumption 2.2, $\phi_0$ defined by (3) can be nonsmooth. Hence, we can smooth $\phi_0$ as follows:

$$\phi_\gamma(u) := \max_{y \in \mathbb{R}^n}\left\{\langle u, Ky\rangle - \psi(y) - \gamma b(y)\right\}, \tag{9}$$

where $b : \mathrm{dom}(\psi) \to \mathbb{R}_+$ is a continuously differentiable and 1-strongly convex function such that $\min_y b(y) = 0$, and $\gamma > 0$ is a smoothness parameter. For example, we can choose $b(y) := \frac{1}{2}\|y - \dot{y}\|^2$ for a fixed $\dot{y}$ or $b(y) := \log(n) + \sum_{j=1}^n y_j \log(y_j)$ defined on a standard simplex $\Delta_n$ [24]. Under Assumption 2.2, $\phi_\gamma$ possesses some useful properties as stated in Lemma A.1 (Supp. Doc. A.1).

Let $y_\gamma^*(u)$ be an optimal solution of the maximization problem in (9), which always exists and is unique. In particular, if $b(y) := \frac{1}{2}\|y - \dot{y}\|^2$, then

$$y_\gamma^*(u) := \arg\max_{y \in \mathbb{R}^n}\left\{\langle u, Ky\rangle - \psi(y) - \frac{\gamma}{2}\|y - \dot{y}\|^2\right\} \equiv \mathrm{prox}_{\psi/\gamma}\big(\dot{y} + \gamma^{-1}K^\top u\big). \tag{10}$$

Hence, when $\psi$ is proximally tractable (i.e., its proximal operator can be computed in a closed-form or with a low-order polynomial time algorithm), computing $y_\gamma^*(u)$ reduces to evaluating the proximal operator of $\psi$ as opposed to solving a complex subproblem as in prox-linear methods [34, 45].

Given $\phi_\gamma$ defined by (9), we consider the following functions:

$$\Phi_\gamma(x) := \phi_\gamma(F(x)) = \phi_\gamma\big(\mathbb{E}_\xi\big[\mathbf{F}(x, \xi)\big]\big) \quad \text{and} \quad \Psi_\gamma(x) := \Phi_\gamma(x) + \mathcal{R}(x). \tag{11}$$

In this case, under Assumptions 2.1 and 2.2, $\Phi_\gamma$ is continuously differentiable, and

$$\nabla\Phi_\gamma(x) = F'(x)^\top \nabla\phi_\gamma(F(x)) = F'(x)^\top K y_\gamma^*(F(x)). \tag{12}$$

**Smoothness:** Moreover, $\Phi_\gamma(\cdot)$ is $L_{\Phi_\gamma}$-smooth with $L_{\Phi_\gamma} := M_{\phi_\gamma} L_F + M_F^2 L_{\phi_\gamma}$ (see [44]), i.e.:

$$\|\nabla\Phi_\gamma(x) - \nabla\Phi_\gamma(\hat{x})\| \le L_{\Phi_\gamma}\|x - \hat{x}\|, \quad \forall x, \hat{x} \in \text{dom}(\Psi_0), \tag{13}$$

where $M_{\phi_\gamma} := M_\psi \|K\|$ and $L_{\phi_\gamma} := \frac{\|K\|^2}{\gamma + \mu_\psi}$ are given in Lemma A.1.

**Gradient mapping:** Let us recall the following gradient mapping of $\Psi_\gamma(\cdot)$ given in (11) as

$$\mathcal{G}_\eta(x) := \frac{1}{\eta}\left(x - \text{prox}_{\eta\mathcal{R}}(x - \eta\nabla\Phi_\gamma(x))\right), \quad \text{for any } \eta > 0. \tag{14}$$

This mapping will be used to characterize approximate KKT points of (1) in Definition 2.1.

## 3 The proposed algorithm and its convergence analysis

First, we introduce a stochastic estimator for $\nabla\Phi_\gamma$. Then, we develop our main algorithm and analyze its convergence and oracle complexity. Finally, we show how to construct an $\epsilon$-KKT point of (1).

### 3.1 Stochastic estimators and the algorithm

Since $F$ is the expectation of a stochastic function $\mathbf{F}$, we exploit the hybrid stochastic estimators for $F$ and its Jacobian $F'$ introduced in [35]. More precisely, given a sequence $\{x_t\}$ generated by a stochastic algorithm, our hybrid stochastic estimators $\tilde{F}_t$ and $\tilde{J}_t$ are defined as follows:

$$\begin{cases} \tilde{F}_t := \beta_{t-1}\tilde{F}_{t-1} + \frac{\beta_{t-1}}{b_1}\sum_{\xi_i \in \mathcal{B}_t^1}\left[\mathbf{F}(x_t, \xi_i) - \mathbf{F}(x_{t-1}, \xi_i)\right] + \frac{(1-\beta_{t-1})}{b_2}\sum_{\zeta_i \in \mathcal{B}_t^2}\mathbf{F}(x_t, \zeta_i) \\ \tilde{J}_t := \hat{\beta}_{t-1}\tilde{J}_{t-1} + \frac{\hat{\beta}_{t-1}}{\hat{b}_1}\sum_{\hat{\xi}_i \in \hat{\mathcal{B}}_t^1}\left[\mathbf{F}'(x_t, \hat{\xi}_i) - \mathbf{F}'(x_{t-1}, \hat{\xi}_i)\right] + \frac{(1-\hat{\beta}_{t-1})}{\hat{b}_2}\sum_{\hat{\zeta}_i \in \hat{\mathcal{B}}_t^2}\mathbf{F}'(x_t, \hat{\zeta}_i), \end{cases} \tag{15}$$

where $\beta_{t-1}, \hat{\beta}_{t-1} \in [0, 1]$ are given weights, and the initial estimators $\tilde{F}_0$ and $\tilde{J}_0$ are defined as

$$\tilde{F}_0 := \frac{1}{b_0}\sum_{\xi_i \in \mathcal{B}^0}\mathbf{F}(x_0, \xi_i) \quad \text{and} \quad \tilde{J}_0 := \frac{1}{\hat{b}_0}\sum_{\hat{\xi}_i \in \hat{\mathcal{B}}^0}\mathbf{F}'(x_0, \hat{\xi}_i). \tag{16}$$

Here, $\mathcal{B}^0$, $\hat{\mathcal{B}}^0$, $\mathcal{B}_t^1$, $\hat{\mathcal{B}}_t^1$, $\mathcal{B}_t^2$, and $\hat{\mathcal{B}}_t^2$ are mini-batches of sizes $b_0$, $\hat{b}_0$, $b_1$, $\hat{b}_1$, $b_2$, and $\hat{b}_2$, respectively. We allow $\mathcal{B}_t^1$ to be **correlated** with $\mathcal{B}_t^2$, and $\hat{\mathcal{B}}_t^1$ to be **correlated** with $\hat{\mathcal{B}}_t^2$. We also do not require any independence between these mini-batches. When $\mathcal{B}_t^1 \equiv \mathcal{B}_t^2$ and $\hat{\mathcal{B}}_t^1 \equiv \hat{\mathcal{B}}_t^2$, our estimators reduce the STORM estimators studied in [7] as a special case. Clearly, with the choices $\mathcal{B}_t^1 \equiv \mathcal{B}_t^2$ and $\hat{\mathcal{B}}_t^1 \equiv \hat{\mathcal{B}}_t^2$, we can save $b_1$ function value evaluations and $\hat{b}_1$ Jacobian evaluations at each iteration.

For $\tilde{F}_t$ and $\tilde{J}_t$ defined by (15), we introduce a stochastic estimator for the gradient $\nabla\Phi_\gamma(x_t) = F'(x_t)^\top\nabla\phi_\gamma(F(x_t))$ of $\Phi_\gamma(\cdot)$ in (11) at $x_t$ as follows:

$$v_t := \tilde{J}_t^\top\nabla\phi_\gamma(\tilde{F}_t) \equiv \tilde{J}_t^\top K y_\gamma^*(\tilde{F}_t). \tag{17}$$

To evaluate $v_t$, we need to compute $y_\gamma^*(\tilde{F}_t)$, which requires just one $\text{prox}_{\gamma\psi}$ if we use (10). Moreover, due to (16) and (17), evaluating $v_0$ does not require the full matrix $\tilde{J}_0$, but a matrix-vector product $\tilde{J}_0^\top K y_\gamma^*(\tilde{F}_0)$, which is often cheaper than evaluating $\tilde{J}_0$.

Using the new estimator $v_t$ of $\nabla\Phi_\gamma(x_t)$ in (17), we propose Algorithm 1 to solve (1).

Algorithm 1 is designed by adopting the idea in [35], where it can start from two initial batches $\mathcal{B}^0$ and $\hat{\mathcal{B}}^0$ to generate a good approximation for the search direction $v_0$ before getting into the main loop. But if diminishing step-sizes are use, it does not require such initial batchs. However, it has 3 major differences compared to [35]: the dual step $y_{\gamma_t}^*(\tilde{F}_t)$, the estimator $v_t$, and the dynamic parameter updates. Note that, as explained in (10), since the dual step $y_{\gamma_t}^*(\tilde{F}_t)$ can be computed using $\text{prox}_{\gamma\psi}$, Algorithm 1 is single loop, making it easy to implement in practice compared to methods based on SVRG [13] and SARAH [26] such as [21, 43].

### 3.2 Convergence analysis of Algorithm 1

Let $\mathcal{F}_t$ be the $\sigma$-field generated by Algorithm 1 up to the $t$-th iteration, which is defined as follows:

$$\mathcal{F}_t := \sigma\left(x_0, \mathcal{B}^0, \hat{\mathcal{B}}^0, \mathcal{B}_1^1, \hat{\mathcal{B}}_1^1, \mathcal{B}_1^2, \hat{\mathcal{B}}_1^2, \cdots, \mathcal{B}_t^1, \hat{\mathcal{B}}_t^1, \mathcal{B}_t^2, \hat{\mathcal{B}}_t^2\right). \tag{18}$$

---

**Algorithm 1** (Smoothing Hybrid Variance-Reduced SGD Algorithm for solving (1))

1: **Inputs:** An arbitrarily initial point $x_0 \in \mathrm{dom}(\Psi_0)$.

2:     Input $\beta_0, \hat{\beta}_0 \in (0,1)$, $\gamma_0 \geq 0$, $\eta_0 > 0$, and $\theta_0 \in (0,1]$ (specified in Subsection 3.2).

3: **Initialization:** Generate $\tilde{F}_0$ and $\tilde{J}_0$ as in (16) with mini-batch sizes $b_0$ and $\hat{b}_0$, respectively.

4:     Solve (9) to obtain $y_{\gamma_0}^*(\tilde{F}_0)$. Then, evaluate $v_0 := \tilde{J}_0^\top K y_{\gamma_0}^*(\tilde{F}_0)$.

5:     Update $\hat{x}_1 := \mathrm{prox}_{\eta_0 \mathcal{R}}(x_0 - \eta_0 v_0)$ and $x_1 := (1-\theta_0)x_0 + \theta_0 \hat{x}_1$.

6: **For** $t := 1, \cdots, T$ **do**

7:     Construct $\tilde{F}_t$ and $\tilde{J}_t$ as in (15) and $v_t := \tilde{J}_t^\top K y_{\gamma_t}^*(\tilde{F}_t)$, where $y_{\gamma_t}^*(\tilde{F}_t)$ solves (9).

8:     Update $\hat{x}_{t+1} := \mathrm{prox}_{\eta_t \mathcal{R}}(x_t - \eta_t v_t)$ and $x_{t+1} := (1-\theta_t)x_t + \theta_t \hat{x}_{t+1}$.

9:     Update $\beta_{t+1}, \hat{\beta}_{t+1}, \theta_{t+1} \in (0,1)$, $\eta_{t+1} > 0$, and $\gamma_{t+1} \geq 0$ if necessary.

10: **EndFor**

11: **Output:** Choose $\bar{x}_T$ randomly from $\{x_0, x_1, \cdots, x_T\}$ with $\mathbf{Prob}\{\bar{x}_T = x_t\} = \frac{\theta_t / L_{\Phi_{\gamma_t}}}{\sum_{t=0}^T \theta_t / L_{\Phi_{\gamma_t}}}$.

---

If $\psi$ is strongly convex, then, without loss of generality, we can assume $\mu_\psi := 1$. Otherwise, we can rescale it. Moreover, for the sake of our presentation, for a given $c_0 > 0$, we introduce:

$$P := \frac{\sqrt{26}\|K\|}{3\sqrt{c_0}}\sqrt{\kappa M_F^4\|K\|^2 + c_0\hat{\kappa}L_F^2 M_\psi^2}, \quad Q := \frac{26}{9c_0}\|K\|^2\big(\kappa M_F^4\|K\|^2\sigma_F^2 + c_0\hat{\kappa}M_\psi^2\sigma_J^2\big),$$

$$L_{\Phi_0} := L_F M_\psi\|K\| + M_F^2\|K\|^2, \qquad \text{and} \quad L_{\Phi_\gamma} := L_F M_\psi\|K\| + \frac{M_F^2\|K\|^2}{\gamma}, \tag{19}$$

where $\gamma > 0$, $M_F$, $L_F$, $\sigma_F$, and $\sigma_J$ are given in Assumption 2.1 and $M_\psi$ is in Assumption 2.2. Here, $\kappa := 1$ if the mini-batch $\mathcal{B}_t^1$ is independent of $\mathcal{B}_t^2$, and $\kappa := 2$, otherwise. Similarly, $\hat{\kappa} := 1$ if $\hat{\mathcal{B}}_t^1$ is independent of $\hat{\mathcal{B}}_t^2$, and $\hat{\kappa} := 2$, otherwise.

### 3.2.1 The strongly concave case

Theorem 3.1, whose proof is in Supp. Doc. B.3, analyzes convergence rate and complexity of Algorithm 1 for the smooth case of $\phi_0$ in (2) (i.e., $\psi$ is strongly convex).

**Theorem 3.1** (**Constant step-size**). *Suppose that Assumptions 2.1 and 2.2 hold, $\psi$ is $\mu_\psi$-strongly convex with $\mu_\psi := 1$, and $P$, $Q$, and $L_{\Phi_0}$ are defined in (19). Given a mini-batch $0 < b \leq \hat{b}_0(T+1)$, let $b_0 := c_0\hat{b}_0$, $\hat{b}_1 = \hat{b}_2 := b$, and $b_1 = b_2 := c_0 b$. Let $\{x_t\}_{t=0}^T$ be generated by Algorithm 1 using*

$$\gamma_t := 0, \quad \beta_t = \hat{\beta}_t := 1 - \frac{b^{1/2}}{[\hat{b}_0(T+1)]^{1/2}}, \quad \theta_t = \theta := \frac{L_{\Phi_0}b^{3/4}}{P[\hat{b}_0(T+1)]^{1/4}}, \quad \text{and} \quad \eta_t = \eta := \frac{2}{L_{\Phi_0}(3+\theta)}, \tag{20}$$

*provided that $\frac{\hat{b}_0(T+1)}{b^3} > \frac{L_{\Phi_0}^4}{P^4}$. Let $b_0 := c_1^2[b(T+1)]^{1/3}$ for some $c_1 > 0$. Then, we have*

$$\mathbb{E}\big[\|\mathcal{G}_\eta(\bar{x}_T)\|^2\big] \leq \frac{\Delta_0}{[b(T+1)]^{2/3}}, \quad \text{where} \quad \Delta_0 := 16P\sqrt{c_1}\big[\Psi_0(x_0) - \Psi_0^\star\big] + \frac{24Q}{c_1}. \tag{21}$$

*For a given tolerance $\varepsilon > 0$, the total number of iterations $T$ to obtain $\mathbb{E}\big[\|\mathcal{G}_\eta(\bar{x}_T)\|^2\big] \leq \varepsilon^2$ is at most $T := \big\lfloor \frac{\Delta_0^{3/2}}{b\varepsilon^3} \big\rfloor$. The total numbers of function evaluation $\mathbf{F}(x_t, \xi)$ and its Jacobian evaluations $\mathbf{F}'(x_t, \xi)$ are at most $\mathcal{T}_F := \big\lfloor \frac{c_0 c_1^2 \Delta_0^{1/2}}{\varepsilon} + \frac{3c_0 \Delta_0^{3/2}}{\varepsilon^3} \big\rfloor$ and $\mathcal{T}_J := \big\lfloor \frac{c_1^2 \Delta_0^{1/2}}{\varepsilon} + \frac{3\Delta_0^{3/2}}{\varepsilon^3} \big\rfloor$, respectively.*

Theorem 3.2 states convergence of Algorithm 1 using diminishing step-size (see Supp. Doc. B.4).

**Theorem 3.2** (**Diminishing step-size**). *Suppose that Assumptions 2.1 and 2.2 hold, $\psi$ is $\mu_\psi$-strongly convex with $\mu_\psi := 1$ (i.e., $\phi_0$ in (2) is smooth). Let $\{x_t\}_{t=0}^T$ be generated by Algorithm 1 using the mini-batch sizes as in Theorem 3.1, and increasing weight and diminishing step-sizes as*

$$\gamma_t := 0, \quad \beta_t = \hat{\beta}_t := 1 - \frac{1}{(t+2)^{2/3}}, \quad \theta_t := \frac{L_{\Phi_0}\sqrt{b}}{P(t+2)^{1/3}}, \quad \text{and} \quad \eta_t := \frac{2}{L_{\Phi_0}(3+\theta_t)}. \tag{22}$$

*Then, for all $T \geq 0$, and $(\bar{x}_T, \bar{\eta}_T)$ chosen as $\mathbf{Prob}\big\{\mathcal{G}_{\bar{\eta}_T}(\bar{x}_T) = \mathcal{G}_{\eta_t}(x_t)\big\} = \frac{\theta_t}{\sum_{t=0}^T \theta_t}$, we have*

$$\mathbb{E}\big[\|\mathcal{G}_{\bar{\eta}_T}(\bar{x}_T)\|^2\big] \leq \frac{32P[\Psi_0(x_0) - \Psi_0^\star]}{3\sqrt{b}\big[(T+3)^{2/3} - 2^{2/3}\big]} + \frac{32Q}{3\big[(T+3)^{2/3} - 2^{2/3}\big]}\left[\frac{2^{1/3}}{\hat{b}_0} + \frac{2(1+\log(T+1))}{b}\right]. \tag{23}$$

If we set $b = \hat{b}_0 = 1$, then our convergence rate is $\mathcal{O}\left(\frac{\log(T)}{T^{2/3}}\right)$ with a $\log(T)$ factor slower than (21).

However, it does not require a large initial mini-batch $\hat{b}_0$ as in Theorem 3.1. In Theorems 3.1 and 3.2, we do not need to smooth $\phi_0$. Hence, $\gamma_t$ is absent in Algorithm 1, i.e., $\gamma_t = 0$ for $t \geq 0$.

### 3.2.2 The non-strongly concave case

Now, we consider the case $\mu_\psi = 0$, i.e., $\psi$ is non-strongly convex (or equivalently, (1) is non-strongly concave in $y$), leading to the nonsmoothness of $\phi_0$ in (2). Theorem 3.3 states convergence of Algorithm 1 in this case, whose proof is in Supp. Doc. B.5.

**Theorem 3.3** (**Constant step-size**). *Assume that Assumptions 2.1 and 2.2 hold, $\psi$ in (1) is non-strongly convex (i.e., $\phi_0$ is nonsmooth), and $P$, $Q$, and $L_{\Phi_\gamma}$ are defined in (19). Let $b$ and $\hat{b}_0$ be two positive integers, $c_0 > 0$, and $\{x_t\}_{t=0}^T$ be generated by Algorithm 1 after $T$ iterations using:*

$$
\begin{cases}
\hat{b}_1 = \hat{b}_2 := b, & b_1 = b_2 := \frac{c_0 b}{\gamma^2}, & \hat{b}_0 := c_1^2[b(T+1)]^{1/3}, & b_0 := \frac{c_0 \hat{b}_0}{\gamma^2}, & \gamma_t := \gamma \in (0,1], \\
\beta_t = \hat{\beta}_t = 1 - \frac{b^{1/2}}{[\hat{b}_0(T+1)]^{1/2}}, & \theta_t = \theta := \frac{L_{\Phi_\gamma} b^{3/4}}{P[\hat{b}_0(T+1)]^{1/4}}, & and & \eta_t = \eta := \frac{2}{L_{\Phi_\gamma}(3+\theta)}.
\end{cases}
\tag{24}
$$

*Then, with $B_\psi$ defined in Lemma A.1, the following bound holds*

$$
\mathbb{E}\big[\|\mathcal{G}_\eta(\bar{x}_T)\|^2\big] \leq \frac{\hat{\Delta}_0}{[b(T+1)]^{2/3}}, \quad where \quad \hat{\Delta}_0 := 16\sqrt{c_1}P\big(\Psi_0(x_0) - \Psi_0^\star + B_\psi\big) + \frac{24Q}{c_1}. \tag{25}
$$

*The total number of iterations $T$ to achieve $\mathbb{E}\big[\|\mathcal{G}_\eta(\bar{x}_T)\|^2\big] \leq \varepsilon^2$ is at most $T := \big\lfloor \frac{\hat{\Delta}_0^{3/2}}{b\varepsilon^3} \big\rfloor$. The total numbers of function evaluations $\mathcal{T}_F$ and Jacobian evaluations $\mathcal{T}_J$ are respectively at most*

$$
\mathcal{T}_F := \frac{c_0 \hat{\Delta}_0^{1/2}}{\gamma^2 \varepsilon} + \frac{3c_0 \hat{\Delta}_0^{3/2}}{\gamma^2 \varepsilon^3} = \mathcal{O}\Big(\frac{\hat{\Delta}_0^{3/2}}{\gamma^2 \varepsilon^3}\Big) \quad and \quad \mathcal{T}_J := \frac{\hat{\Delta}_0^{1/2}}{\varepsilon} + \frac{3\hat{\Delta}_0^{3/2}}{\varepsilon^3} = \mathcal{O}\Big(\frac{\hat{\Delta}_0^{1.5}}{\varepsilon^3}\Big).
$$

*If we choose $\gamma := c_2 \varepsilon$ for some $c_2 > 0$, then $\mathcal{T}_F = \frac{c_0 \hat{\Delta}_0^{1/2}}{c_2^2 \varepsilon^3} + \frac{3c_0 \hat{\Delta}_0^{3/2}}{c_2^2 \varepsilon^5} = \mathcal{O}\big(\frac{\hat{\Delta}_0^{3/2}}{\varepsilon^5}\big)$.*

Alternatively, we can also establish convergence and estimate the complexity of Algorithm 1 with diminishing step-size in Theorem 3.4, whose proof is in Supp. Doc. B.6.

**Theorem 3.4** (**Diminishing step-size**). *Suppose that Assumptions 2.1 and 2.2 hold, $\psi$ is non-strongly convex (i.e., $\phi_0$ is possibly nonsmooth), and $P$, $Q$, $L_{\Phi_{\gamma_t}}$ are defined by (19). Given mini-batch sizes $b > 0$ and $\hat{b}_0 > 0$, let $b_0 := \frac{c_0 \hat{b}_0}{\gamma_0^2}$, $b_1^t = b_2^t := \frac{c_0 b}{\gamma_t^2}$, and $\hat{b}_1 = \hat{b}_2 := b$ for some $c_0 > 0$. Let $\{x_t\}_{t=0}^T$ be generated by Algorithm 1 using increasing weight and diminishing step-sizes as*

$$
\gamma_t := \frac{1}{(t+2)^{1/3}}, \quad \beta_t = \hat{\beta}_t := 1 - \frac{1}{(t+2)^{2/3}}, \quad \theta_t := \frac{L_{\Phi_{\gamma_t}} b^{1/2}}{P(t+2)^{1/3}}, \quad and \quad \eta_t := \frac{2}{L_{\Phi_{\gamma_t}}(3+\theta_t)}. \tag{26}
$$

*For $(\bar{x}_T, \bar{\eta}_T)$ chosen as $\mathbf{Prob}\big\{\mathcal{G}_{\bar{\eta}_T}(\bar{x}_T) = \mathcal{G}_{\eta_t}(x_t)\big\} = \big[\sum_{t=0}^T (\theta_t/L_{\Phi_{\gamma_t}})\big]^{-1}(\theta_t/L_{\Phi_{\gamma_t}})$, we have*

$$
\begin{aligned}
\mathbb{E}\big[\|\mathcal{G}_{\bar{\eta}_T}(\bar{x}_T)\|^2\big] \quad \leq \quad & \frac{32P}{3\sqrt{b}[(T+3)^{2/3} - 2^{2/3}]}\Big(\Psi_0(x_0) - \Psi_0^\star + \frac{B_\psi}{(T+2)^{1/3}}\Big) \\
& + \frac{16Q}{3[(T+3)^{2/3} - 2^{2/3}]}\Big(\frac{2^{1/3}}{\hat{b}_0} + \frac{2(1+\log(T+1))}{b}\Big) = \mathcal{O}\Big(\frac{\log(T)}{T^{2/3}}\Big).
\end{aligned}
\tag{27}
$$

Note that since $\gamma_t := \frac{1}{(t+2)^{1/3}}$ (diminishing) and $b_1^t = b_2^t := \frac{c_0 b}{\gamma_t^2}$, we have $b_1^t = b_2^t = c_0 b(t+2)^{2/3}$, which shows that the mini-batch sizes of the function estimation $\tilde{F}_t$ are chosen in increasing manner (not fixed at a large size for all $t$), which can save computational cost for $F$. The batch sizes $b$ and $\hat{b}_0$ in Theorems 3.3 and 3.4 must be chosen to guarantee $\beta_t, \theta_t \in (0,1]$.

### 3.3 Constructing approximate KKT point for (1) from Algorithm 1

Existing works such as [21, 43, 45] do not show how to construct an $\epsilon$-KKT point of (1) or an $\epsilon$-stationary point of (2) from $\bar{x}_T$ with $\mathbb{E}\big[\|\mathcal{G}_{\bar{\eta}_T}(\bar{x}_T)\|^2\big] \leq \varepsilon^2$. Lemma 3.1, whose proof is in Supp. Doc. A.3, shows one way to construct an $\epsilon$-KKT point of (1) in the sense of Definition 2.1 with $\epsilon := \mathcal{O}(\varepsilon)$ from the output $\bar{x}_T$ of Theorems 3.1, 3.2, 3.3, and 3.4.

**Lemma 3.1.** *Let $\bar{x}_T$ be computed by Algorithm 1 up to an accuracy $\varepsilon > 0$ after $T$ iterations. Assume that we can approximate $F'(\bar{x}_T)$, $F(\bar{x}_T)$, and $F(\tilde{x}^*_{\gamma_T})$, respectively such that*

$$\mathbb{E}\big[\|\tilde{F}(\bar{x}_T) - F(\bar{x}_T)\|\big] \leq (\mu_\psi + \gamma_T)\varepsilon, \quad \mathbb{E}\big[\|(\tilde{J}(\bar{x}_T) - F'(\bar{x}_T))^\top \nabla\phi_{\gamma_T}(\tilde{F}(\bar{x}_T))\|\big] \leq \varepsilon, \tag{28}$$

*and* $\mathbb{E}\big[\|\tilde{F}(\tilde{x}^*_{\gamma_T}) - F(\tilde{x}^*_{\gamma_T})\|\big] \leq \varepsilon.$

*Let us denote $\widetilde{\nabla}\Phi_{\gamma_T}(\bar{x}_T) := \tilde{J}(\bar{x}_T)^\top \nabla\phi_\gamma(\tilde{F}(\bar{x}_T))$ and compute $(\tilde{x}^*_{\gamma_T}, \tilde{y}^*_{\gamma_T})$ as*

$$\tilde{x}^*_{\gamma_T} := \mathrm{prox}_{\bar{\eta}_T \mathcal{R}}(\bar{x}_T - \bar{\eta}_T \widetilde{\nabla}\Phi_{\gamma_T}(\bar{x}_T)) \quad \text{and} \quad \tilde{y}^*_{\gamma_T} := y^*_{\gamma_T}(\tilde{F}(\tilde{x}^*_{\gamma_T})) \text{ by } (9). \tag{29}$$

*Suppose that $\mathbb{E}\big[\|\mathcal{G}_{\bar{\eta}_T}(\bar{x}_T)\|^2\big] \leq \varepsilon^2$ and $0 \leq \gamma_T \leq c_2\varepsilon$ for a constant $c_2 \geq 0$. Then*

$$\mathbb{E}\big[\mathcal{E}(\tilde{x}^*_{\gamma_T}, \tilde{y}^*_{\gamma_T})\big] \leq \epsilon, \quad \text{where} \quad \epsilon := \big[\tfrac{13}{3} + \tfrac{8}{3}M_F\|K\|^2 + c_2 D_\psi\big]\varepsilon, \tag{30}$$

*where $D_\psi$ is in Lemma A.1 and $\mathcal{E}(\cdot)$ is given by (8). In other words, $(\tilde{x}^*_{\gamma_T}, \tilde{y}^*_{\gamma_T})$ is an $\epsilon$-KKT of (1).*

If we use stochastic estimators as in (16) to form $\tilde{F}(\bar{x}_T)$, $\tilde{J}(\bar{x}_T)$, and $\tilde{F}(\tilde{x}^*_{\gamma_T})$ with batch sizes $b_T$, $\hat{b}_T$, and $\tilde{b}_T$, respectively, then (28) holds if we choose $b_T := \big\lfloor \frac{\sigma_F^2}{(\mu_\psi + \gamma_T)^2 \varepsilon^2} \big\rfloor$, $\hat{b}_T := \big\lfloor \frac{\sigma_J^2}{\varepsilon^2} \big\rfloor$, and $\tilde{b}_T := \big\lfloor \frac{\sigma_F^2}{\varepsilon^2} \big\rfloor$. We do not explicitly compute Jacobian $\tilde{J}(\bar{x}_T)$, but its matrix-vector product $\tilde{J}(\bar{x}_T)^\top \nabla\phi_{\gamma_T}(\tilde{F}(\bar{x}_T))$. This extra cost is dominated by $\mathcal{T}_J$ and $\mathcal{T}_F$ in Theorems 3.1, 3.2, 3.3, and 3.4. For $\bar{x}_T$ computed by Theorems 3.1 and 3.2, we can set $\gamma_T := 0$, or equivalently, $c_2 := 0$. For $\bar{x}_T$ computed by Theorem 3.3, since $\gamma_T := c_2\varepsilon$ and $\mu_\psi = 0$, we have $b_T = \big\lfloor \frac{\sigma_F^2}{c_2^2 \varepsilon^4} \big\rfloor < \mathcal{T}_F = \mathcal{O}\big(\frac{\hat{\Delta}_0^{3/2}}{\varepsilon^5}\big)$.

## 4 Numerical experiments

We use two examples to illustrate our algorithm and compare it with existing methods. Our code is implemented in Python 3.6.3, running on a Linux desktop (3.6GHz Intel Core i7 and 16Gb memory).

### 4.1 Risk-averse portfolio optimization

We consider a risk-averse portfolio optimization problem studied in [22], and recent used in [44]:

$$\max_{x \in \mathbb{R}^p} \Big\{ \mathbb{E}_\xi\big[h_\xi(x)\big] - \rho\mathrm{Var}_\xi\big[h_\xi(x)\big] \equiv \mathbb{E}_\xi\big[h_\xi(x)\big] + \rho\mathbb{E}_\xi\big[h_\xi(x)\big]^2 - \rho\mathbb{E}_\xi\big[h_\xi^2(x)\big] \Big\}, \tag{31}$$

where $\rho > 0$ is a trade-off parameter and $h_\xi(x)$ is a reward for the portfolio vector $x$. Following [44], (31) can be reformulated into (2), where $\phi_0(u) = u_1 + \rho u_1^2 - \rho u_2$ is smooth, and $\mathbf{F}(x, \xi) = (h_\xi(x), h_\xi^2(x))^\top$. Suppose further that we only consider $N$ periods of time. Then we can view $\xi \in \{1, \cdots, N\}$ as a discrete random variable and define $h_i(x) := \langle r_i, x \rangle$ as a linear reward function, where $r_i := (r_{i1}, \cdots, r_{ip})^\top$ and $r_{ij}$ represents the return per unit of $j$ at time $i$. We also choose $\mathcal{R}(x) := \lambda\|x\|_1$ as a regularizer to promote sparsity as in [44].

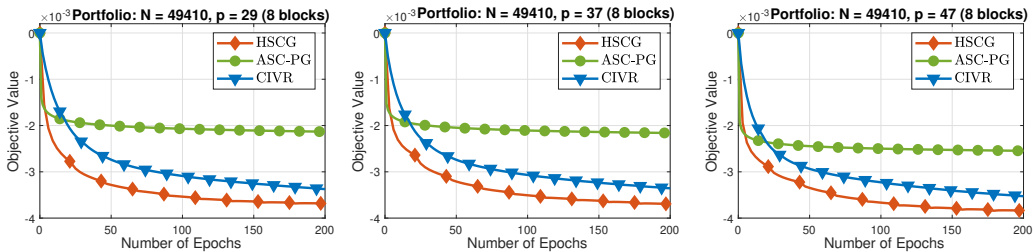

Figure 1: Comparison of three algorithms for solving (31) on 3 different datasets.

We implement our algorithm, abbreviated by HSCG (i.e., Hybrid Stochastic Compositional Gradient for short), and test it on three real-world portfolio datasets, which contain 29, 37, and 47 portfolios, respectively, from the Keneth R. French Data Library [1]. We set $\rho := 0.2$ and $\lambda := 0.01$ as in [44]. For comparison, we also implement 2 methods, called CIVR in [44] and ASC-PG in [38]. The step-size

$\eta$ of all algorithms are well tuned from a set of trials $\{1, 0.5, 0.1, 0.05, 0.01, 0.001, 0.0001\}$. The performance of 3 algorithms are shown in Figure 1 for three datasets using $b := \lfloor N/8 \rfloor$ (8 blocks).

One can observe from Fig. 1 that both HSCG and CIVR highly outperform ASC-PG due to their variance-reduced property. HSCG is slightly better than CIVR since it has a flexible step-size $\theta_t$. Note that, in theory, CIVR requires a large batch for both function values and Jacobian, which may affect its performance, while HSCG can work with a wide range of batches, including singe sample.

## 4.2 Stochastic minimax problem

We consider the following regularized stochastic minimax problem studied, e.g., in [31]:

$$\min_{x \in \mathbb{R}^p} \left\{ \max_{1 \le i \le m} \{\mathbb{E}_\xi[\mathbf{F}_i(x, \xi)]\} + \tfrac{\lambda}{2}\|x\|^2 \right\}, \tag{32}$$

where $\mathbf{F}_i : \mathbb{R}^p \times \Omega \to \mathbb{R}_+$ can be taken as the loss function of the $i$-th model. If we define $\phi_0(u) := \max_{1 \le i \le m}\{u_i\}$ and $\mathcal{R}(x) := \tfrac{\lambda}{2}\|x\|^2$, then (32) can be reformulated into (2). Since $u_i \ge 0$, we have $\phi_0(u) := \max_{1 \le i \le m}\{u_i\} = \|u\|_\infty = \max_{\|y\|_1 \le 1}\{\langle u, y \rangle\}$, which is nonsmooth. Therefore, we can smooth $\phi_0$ as $\phi_\gamma(u) := \max_{\|y\|_1 \le 1}\{\langle u, y \rangle - (\gamma/2)\|y\|^2\}$ using $b(y) := \tfrac{1}{2}\|y\|^2$.

In this example, we employ (32) to solve a model selection problem in binary classification with nonconvex loss, see, e.g., [46]. Suppose that we have four ($m = 4$) different nonconvex losses: $\mathbf{F}_1(x, \xi) := 1 - \tanh(b\langle a, x \rangle)$, $\mathbf{F}_2(x, \xi) := \log(1 + \exp(-b\langle a, x \rangle)) - \log(1 + \exp(-b\langle a, x \rangle - 1))$, $\mathbf{F}_3(x, \xi) := (1 - 1/(\exp(-b\langle a, x \rangle) + 1))^2$, and $\mathbf{F}_4(x, \xi) := \log(1 + \exp(-b\langle a, x \rangle))$ (see [46] for more details), where $\xi := (a, b)$ represents examples. We assume that we have $N$ examples of $\xi$.

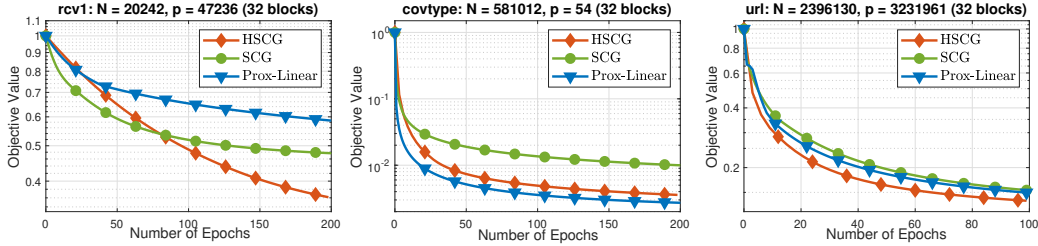

Figure 2: Comparison of three algorithms for solving (32) on 3 different datasets.

We implement three algorithms: HSCG, SCG in [37], and Prox-Linear in [45]. We test them on 3 datasets from LIBSVM [6]. We set $\lambda := 10^{-4}$ and update our $\gamma_t$ parameter as $\gamma_t := \frac{1}{2(t+1)^{1/3}}$. The step-size $\eta$ of all algorithms are well tuned from $\{1, 0.5, 0.1, 0.05, 0.01, 0.001, 0.0001\}$, and their performance is shown in Figure 2 for three datasets: **rcv1**, **covtype**, and **url** with 32 blocks.

One can observe from Figure 2 that HSCG outperforms SCG and Prox-Linear on **rcv1** and **url**. For **covtype**, since $p$ is very small, allowing us to evaluate the prox-linear operator to a high accuracy, Prox-Linear slightly performs better than ours and much better than SCG. Note that solving the subproblem of Prox-Linear is expensive when $p$ is large. Hence, if $p$ is large, Prox-Linear becomes much slower than HSCG and SCG in terms of time. Due to space limit, we refer to Supp. Doc. D for further details of experiments and additional results.

## 5 Conclusions

We have proposed a new single loop hybrid variance-reduced SGD algorithm, Algorithm 1, to solve a class of nonconvex-concave saddle-point problems. The main idea is to combine both smoothing idea [24] and hybrid SGD approach in [35] to develop novel algorithms with less tuning effort. Our algorithm relies on standard assumptions, and can achieve the best-known oracle complexity, and in some cases, the optimal oracle complexity. It also has several computational advantages compared to existing methods such as avoiding expensive subproblems, working with both single sample and mini-batches, and using constant and diminishing step-sizes. We have also proposed a simple restarting variant, Algorithm 2, in Supp. Doc. C to improve practical performance in the constant step-size case without sacrificing complexity bounds. We believe that both algorithms and theoretical results are new, even in the smooth case, compared to [34, 43, 45].

## 6  Broader Impact

This work could potentially have positive impact in different fields where nonconvex-concave minimax and nonconvex compositional optimization models as (1) and (2) are used. For instance, robust learning, distributionally robust optimization, zero-sum game, and generative adversarial nets (GANs) applications are concrete examples under certain settings. We emphasize that the nonconvex-concave minimax problem (1) studied in this paper remains challenging to solve for global solutions. Existing methods can only find an approximate KKT (Karush-Kuhn-Tucker) point in general. This paper proposed new algorithms to tackle a class of nonconvex-concave minimax problems, but they can only guarantee to find an approximate KKT point, which may not be an approximate global solution of the model. This could lead to a negative impact if one expects to find an approximate global solution instead of an approximate KKT point without further investigation. Apart from the above impact, since this paper is a theoretical work, it does not present any other foreseeable societal consequence.

## Acknowledgments and Disclosure of Funding

The work of Quoc Tran-Dinh and Deyi Liu is partially supported by the National Science Foundation (NSF), grant no. DMS-1619884, and the Office of Naval Research (ONR), grant No. N00014-20-1-2088. The authors would also like to thank all the anonymous reviewers and the ACs for their constructive comments to improve the paper.

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
