[Supplementary Material]



# Hybrid Variance-Reduced SGD Algorithms For Minimax Problems with Nonconvex-Linear Function

**Quoc Tran-Dinh**[*]        **Deyi Liu**[*]        **Lam M. Nguyen**[†]

[*]Department of Statistics and Operations Research
The University of North Carolina at Chapel Hill, Chapel Hill, NC 27599
Emails: {quoctd@email.unc.edu, deyi.liu@live.unc.edu}
[†]IBM Research, Thomas J. Watson Research Center
Yorktown Heights, NY10598, USA.
Email: lamnguyen.mltd@ibm.com

## A    Some technical results and proof of Lemma 3.1

In this Supp. Doc., we provide some useful properties of $\phi_0$ in (3) and its smoothed approximation $\phi_\gamma$ defined by (9) in Section 2. Then we recall and prove some bounds of variance for $\tilde{F}_t$, $\tilde{J}_t$, and $v_t$. Finally, we prove Lemma 3.1 in the main text.

### A.1    Properties of the smoothed function $\phi_\gamma$

Under Assumption 2.2, $\phi_0$ in (3) and $\phi_\gamma$ defined by (9) have the following properties.

**Lemma A.1.**  *Let $\phi_0$ be defined by* (3) *and $\phi_\gamma$ be defined by* (9). *Then, the following statements hold:*

(a)  dom$(\psi)$ *is bounded by $M_\psi$ iff $\phi_0$ is $M_{\phi_0}$-Lipschitz continuous with $M_{\phi_0} := M_\psi\|K\|$.*
(b)  dom$(\psi)$ *is bounded by $M_\psi$ iff $\phi_\gamma$ is Lipschitz continuous with $M_{\phi_\gamma} := M_\psi\|K\|$.*
(c)  *$\phi_\gamma$ is convex and $L_{\phi_\gamma}$-smooth with $L_{\phi_\gamma} := \frac{\|K\|^2}{\gamma+\mu_\psi}$.*
(d)  *It holds that $\phi_\gamma(u) \leq \phi_0(u) \leq \phi_\gamma(u) + \gamma B_\psi$ for all $u \in \mathbb{R}^q$, where $\gamma > 0$ and $B_\psi :=$ sup $\{b(y) \mid y \in$ dom$(\psi)\}$. In addition, we have $D_\psi := \max_{v\in\text{dom}(\psi)} \|\nabla b(v)\| < +\infty$.*
(e)  *We have $\phi_\gamma(u) \leq \phi_{\hat\gamma}(u) + (\hat\gamma - \gamma)b(y^*_\gamma(u)) \leq \phi_{\hat\gamma}(u) + (\hat\gamma - \gamma)B_\psi$ for all $\hat\gamma \geq \gamma > 0$.*

*Proof.*  The statement (a) can be found in [3, Corollary 17.19].

Since $\nabla\phi_\gamma(u) = Ky^*_\gamma(u)$ with $y^*_\gamma(u) \in$ dom$(\psi)$, we have $\|\nabla\phi_\gamma(u)\| \leq \|K\|\|y^*_\gamma(u)\| \leq M_\psi\|K\|$. Applying again [3, Corollary 17.19] we prove (b).

The statement (c) holds due to the well-known Baillon-Haddad theorem [3, Corollary 18.17].

The proof of the first part of (d) can be found in [24]. Under Assumption 2.2 and the continuous differentiability of $b$, we have $D_\psi := \max_{v\in\text{dom}(\psi)} \|\nabla b(v)\| < +\infty$.

Finally, for any $u$ and $y$, since $s(\gamma; u, y) := \langle u, Ky\rangle - \psi(y) - \gamma b(y)$ is linear in $\gamma$. Therefore, $\phi_\gamma(u) := \max_{y\in\mathbb{R}^n} s(\gamma; u, y)$ is convex in $\gamma$ and $\frac{d}{d\gamma}\phi_\gamma(u) = -b(y^*_\gamma(u)) \leq 0$. Consequently, we have $\phi_\gamma(u) + \frac{d}{d\gamma}\phi_\gamma(u)(\hat\gamma - \gamma) = \phi_\gamma(u) - (\hat\gamma - \gamma)b(y^*_\gamma(u)) \leq \phi_{\hat\gamma}(u)$, which implies (e).    $\square$

One common example of $\psi$ in Assumption 2.2 is $\psi(x) := \delta_{\mathcal{X}}(x)$, the indicator of a nonempty, closed, bounded, and convex set $\mathcal{X}$. For instance, $\mathcal{X} := \{y \in \mathbb{R}^n \mid \|y\|_* \leq 1\}$ is a unit ball in the dual norm $\|\cdot\|_*$ of $\|\cdot\|$. Then, we have $\phi_0(u) := \|u\|$, which is clearly Lipschitz continuous. In particular, if $\mathcal{X} := \{y \in \mathbb{R}^n \mid \|y\|_\infty \leq 1\}$, then $\phi_0(u) := \|u\|_1$.

### A.2    Key bounds on the variance of estimators

Next, we provide some useful bounds for the estimators $\tilde{F}_t$ and $\tilde{J}_t$ defined in (15). The following lemma can be found in [35], where we have used the inequality $2\mathbb{E}[\langle a, b\rangle] \leq \mathbb{E}[\|a\|^2] + \mathbb{E}[\|b\|^2]$ in the proof, when $a$ and $b$ are not independent.

**Lemma A.2.** *Let $\tilde{F}_t$ and $\tilde{J}_t$ be defined by* (15)*, and $\mathcal{F}_t$ be defined by* (18)*. Then*

$$
\begin{aligned}
\mathbb{E}_{(\mathcal{B}_t^1, \mathcal{B}_t^2)}\big[\|\tilde{F}_t - F(x_t)\|^2\big] &\leq \beta_{t-1}^2 \|\tilde{F}_{t-1} - F(x_{t-1})\|^2 - \beta_{t-1}^2 \|F(x_t) - F(x_{t-1})\|^2 \\
&\quad + \kappa(1 - \beta_{t-1})^2 \mathbb{E}_{\mathcal{B}_t^2}\big[\|\mathbf{F}(x_t, \zeta_t) - F(x_t)\|^2\big] \\
&\quad + \tfrac{\kappa \beta_{t-1}^2}{b_1} \mathbb{E}_\xi\big[\|\mathbf{F}(x_t, \xi) - \mathbf{F}(x_{t-1}, \xi)\|^2\big], \\
\mathbb{E}_{(\hat{\mathcal{B}}_t^1, \hat{\mathcal{B}}_t^2)}\big[\|\tilde{J}_t - F'(x_t)\|^2\big] &\leq \hat{\beta}_{t-1}^2 \|\tilde{J}_{t-1} - F'(x_{t-1})\|^2 \\
&\quad + \hat{\kappa}(1 - \hat{\beta}_{t-1})^2 \mathbb{E}_{\hat{\mathcal{B}}_t^2}\big[\|\mathbf{F}'(x_t, \hat{\zeta}_t) - F'(x_t)\|^2\big] \\
&\quad + \tfrac{\hat{\kappa} \hat{\beta}_{t-1}^2}{\hat{b}_1} \mathbb{E}_{\hat{\xi}}\big[\|\mathbf{F}'(x_t, \hat{\xi}) - \mathbf{F}'(x_{t-1}, \hat{\xi})\|^2\big].
\end{aligned}
\tag{33}
$$

*Here, $\kappa = 1$ if $\mathcal{B}_t^1$ is independent of $\mathcal{B}_t^2$, and $\kappa = 2$, otherwise. Similarly, $\hat{\kappa} = 1$ if $\hat{\mathcal{B}}_t^1$ is independent of $\hat{\mathcal{B}}_t^2$, and $\hat{\kappa} = 2$, otherwise.*

Furthermore, we can bound the variance of the estimator $v_t$ of $\nabla\Phi_{\gamma_t}(x_t)$ defined in (17) as follows.

**Lemma A.3.** *Let $\Phi_\gamma$ and $v_t$ be defined by* (11) *and* (17)*, respectively. Then, under Assumptions 2.1 and 2.2, we have*

$$
\mathbb{E}\big[\|v_t - \nabla\Phi_{\gamma_t}(x_t)\|^2\big] \leq 2M_F^2 L_{\phi_{\gamma_t}}^2 \mathbb{E}\big[\|\tilde{F}_t - F(x_t)\|^2\big] + 2M_{\phi_{\gamma_t}}^2 \mathbb{E}\big[\|\tilde{J}_t - F'(x_t)\|^2\big].
\tag{34}
$$

*Proof.* First, by the composition rule of derivatives, we can derive

$$
\begin{aligned}
\|v_t - \nabla\Phi_{\gamma_t}(x_t)\|^2 &= \|\tilde{J}_t^\top \nabla\phi_{\gamma_t}(\tilde{F}_t) - F'(x_t)^\top \nabla\phi_{\gamma_t}(F(x_t))\|^2 \\
&= \big\|\tilde{J}_t^\top \nabla\phi_{\gamma_t}(\tilde{F}_t) - F'(x_t)^\top \nabla\phi_{\gamma_t}(\tilde{F}_t) + F'(x_t)^\top \nabla\phi_{\gamma_t}(\tilde{F}_t) \\
&\quad - F'(x_t)^\top \nabla\phi_{\gamma_t}(F(x_t))\big\|^2 \\
&\overset{(i)}{\leq} 2\|(\tilde{J}_t - F'(x_t))^\top \nabla\phi_{\gamma_t}(\tilde{F}_t)\|^2 + 2\|F'(x_t)^\top (\nabla\phi_{\gamma_t}(\tilde{F}_t) - \nabla\phi_{\gamma_t}(F(x_t)))\|^2 \\
&\leq 2\|\nabla\phi_{\gamma_t}(\tilde{F}_t)\|^2 \|\tilde{J}_t - F'(x_t)\|^2 + 2\|\nabla\phi_{\gamma_t}(\tilde{F}_t) - \nabla\phi_{\gamma_t}(F(x_t))\|^2 \|F'(x_t)\|^2 \\
&\overset{(ii)}{\leq} 2M_{\phi_{\gamma_t}}^2 \|\tilde{J}_t - F'(x_t)\|^2 + 2L_{\phi_{\gamma_t}}^2 M_F^2 \|\tilde{F}_t - F(x_t)\|^2.
\end{aligned}
$$

Here, we use $\|a + b\|^2 \leq 2\|a\|^2 + 2\|b\|^2$ in *(i)* and the $M_{\phi_{\gamma_t}}$-Lipschitz continuity, $L_{\phi_{\gamma_t}}$-smoothness of $\phi_{\gamma_t}$, and (5) in *(ii)*. Taking expectation over $\mathcal{F}_{t+1}$ on both sides the last inequality, we obtain

$$
\mathbb{E}\big[\|v_t - \nabla\Phi_{\gamma_t}(x_t)\|^2\big] \leq 2M_F^2 L_{\phi_{\gamma_t}}^2 \mathbb{E}\big[\|\tilde{F}_t - F(x_t)\|^2\big] + 2M_{\phi_{\gamma_t}}^2 \mathbb{E}\big[\|\tilde{J}_t - F'(x_t)\|^2\big],
$$

which proves (34). $\qquad\square$

### A.3 The construction of approximate KKT points for (1)

Recall from (11) that $\Phi_\gamma(x) = \phi_\gamma(F(x))$ and $\nabla\Phi_\gamma(x) = F'(x)^\top \nabla\phi_\gamma(F(x))$, where $\phi_\gamma$ is defined by (9). We define a smoothed approximation problem of (2) as follows:

$$
\min_{x \in \mathbb{R}^p} \Big\{ \Psi_\gamma(x) := \Phi_\gamma(x) + \mathcal{R}(x) \equiv \phi_\gamma(F(x)) + \mathcal{R}(x) \Big\}.
\tag{35}
$$

Clearly, if $\gamma = 0$, then (35) reduces to (2). The optimality condition of (35) becomes

$$
0 \in \nabla\Phi_\gamma(x_\gamma^\star) + \partial\mathcal{R}(x_\gamma^\star) \equiv F'(x_\gamma^\star)^\top \nabla\phi_\gamma(F(x_\gamma^\star)) + \partial\mathcal{R}(x_\gamma^\star).
\tag{36}
$$

Here, $x_\gamma^\star$ is called a stationary point of (35). Therefore, an $\varepsilon$-stationary point $\tilde{x}_\gamma^*$ is defined as

$$
\mathbb{E}\big[\mathrm{dist}\big(0, \nabla\Phi_\gamma(\tilde{x}_\gamma^*) + \partial\mathcal{R}(\tilde{x}_\gamma^*)\big)\big] \leq \varepsilon.
\tag{37}
$$

Again, the expectation $\mathbb{E}[\,\cdot\,]$ is taken over all the randomness generated by the model (35) and the algorithm for finding $\tilde{x}_\gamma^*$.

Alternatively, using the definition of $\phi_\gamma$ in (9), problem (35) can be written as

$$\min_{x\in\mathbb{R}^p} \max_{y\in\mathbb{R}^n} \left\{ \mathcal{R}(x) + \langle F(x), Ky \rangle - \psi(y) - \gamma b(y) \right\}. \tag{38}$$

Its optimality condition becomes

$$0 \in \partial\mathcal{R}(x_\gamma^\star) + F'(x_\gamma^\star)Ky_\gamma^\star \quad \text{and} \quad 0 \in K^\top F(x_\gamma^\star) - \partial\psi(y_\gamma^\star) - \gamma\nabla b(y_\gamma^\star). \tag{39}$$

Using the definition of $\mathcal{E}$ in (8), we have

$$\mathcal{E}(x_\gamma^\star, y_\gamma^\star) := \text{dist}\left(0, \partial\mathcal{R}(x_\gamma^\star) + F'(x_\gamma^\star)Ky_\gamma^\star\right) + \text{dist}\left(0, K^\top F(x_\gamma^\star) - \partial\psi(y_\gamma^\star)\right) \leq \gamma D_\psi. \tag{40}$$

Here, we use the fact that $\|\nabla b(y_\gamma^\star)\| \leq D_\psi$ as stated in Lemma A.1.

Given $\bar{x} \in \text{dom}(\Psi_0)$, let $\tilde{F}(\cdot)$ and $\tilde{J}(\cdot)$ be a stochastic approximation of $F(\cdot)$ and $F'(\cdot)$, respectively. We define $(\tilde{x}_\gamma^*, y_\gamma^*)$ as follows:

$$\begin{cases} \tilde{x}_\gamma^* := \text{prox}_{\eta\mathcal{R}}\left(\bar{x} - \eta\widetilde{\nabla}\Phi_\gamma(\bar{x})\right), \quad \text{where} \quad \widetilde{\nabla}\Phi_\gamma(\bar{x}) := \tilde{J}(\bar{x})^\top\nabla\phi_\gamma(\tilde{F}(\bar{x})), \\ \tilde{y}_\gamma^* := y_\gamma^*(\tilde{F}(\tilde{x}_\gamma^*)) \equiv \arg\min_{y\in\mathbb{R}^n}\left\{\langle K^\top\tilde{F}(\tilde{x}_\gamma^*), y\rangle - \psi(y) - \gamma b(y)\right\}, \end{cases} \tag{41}$$

Note that $\tilde{x}_\gamma^*$ only depends on $\bar{x}$, while $\tilde{y}_\gamma^*$ depends on both $\bar{x}$ and $\tilde{x}_\gamma^*$. Hence, we first compute $\tilde{x}_\gamma^*$ and then compute $\tilde{y}_\gamma^*$.

The following lemma provides key estimates to prove Lemma 3.1 in the main text.

**Lemma A.4.** *Under Assumptions 2.1 and 2.2, for given $\bar{x}$ and $\eta > 0$, $\tilde{x}_\gamma^*$ defined by (41) satisfies*

$$\text{dist}\left(0, \nabla\Phi_\gamma(\tilde{x}_\gamma^*) + \partial\mathcal{R}(\tilde{x}_\gamma^*)\right) \leq \left(1 + \eta L_{\Phi_\gamma}\right)\|\mathcal{G}_\eta(\bar{x})\| + (2 + \eta L_{\Phi_\gamma})\|\nabla\Phi_\gamma(\bar{x}) - \widetilde{\nabla}\Phi_\gamma(\bar{x})\|. \tag{42}$$

*Let $(\tilde{x}_\gamma^*, \tilde{y}_\gamma^*)$ be computed by (41), and $\mathcal{E}(x, y)$ be defined by (8). Then, we have*

$$\begin{aligned} \mathcal{E}(\tilde{x}_\gamma^*, \tilde{y}_\gamma^*) \leq{}& \left(1 + \eta L_{\Phi_\gamma}\right)\|\mathcal{G}_\eta(\bar{x})\| + \gamma D_\psi + \|K\|\|F(\tilde{x}_\gamma^*) - \tilde{F}(\tilde{x}_\gamma^*)\| \\ &+ \left(2 + \eta L_{\Phi_\gamma}\right)\left[\|(\tilde{J}(\bar{x}) - F'(\bar{x}))^\top\nabla\phi_\gamma(\tilde{F}(\bar{x})\| + L_{\phi_\gamma}M_F\|\tilde{F}(\bar{x}) - F(\bar{x})\|\right], \end{aligned} \tag{43}$$

*where $D_\psi$ is defined in Lemma A.1.*

*Proof.* From (41), we have $\bar{x} - \eta\widetilde{\nabla}\Phi_\gamma(\bar{x}) \in \tilde{x}_\gamma^* + \eta\partial\mathcal{R}(\tilde{x}_\gamma^*)$, which is equivalent to

$$r_x^* := \frac{1}{\eta}(\bar{x} - \tilde{x}_\gamma^*) + \left(\nabla\Phi_\gamma(\tilde{x}_\gamma^*) - \widetilde{\nabla}\Phi_\gamma(\bar{x})\right) \in \nabla\Phi_\gamma(\tilde{x}_\gamma^*) + \partial\mathcal{R}(\tilde{x}_\gamma^*). \tag{44}$$

We can bound $r_x^*$ in (44) as follows:

$$\begin{aligned} \|r_x^*\| &\leq \frac{1}{\eta}\|\bar{x} - \tilde{x}_\gamma^*\| + \|\nabla\Phi_\gamma(\tilde{x}_\gamma^*) - \nabla\Phi_\gamma(\bar{x})\| + \|\nabla\Phi_\gamma(\bar{x}) - \widetilde{\nabla}\Phi_\gamma(\bar{x})\| \\ &\leq \frac{1}{\eta}\left(1 + \eta L_{\Phi_\gamma}\right)\|\tilde{x}_\gamma^* - \bar{x}\| + \|\nabla\Phi_\gamma(\bar{x}) - \widetilde{\nabla}\Phi_\gamma(\bar{x})\|. \end{aligned} \tag{45}$$

Next, from (14), let us define $\bar{x}_\gamma^* := \bar{x} - \eta\mathcal{G}_\eta(\bar{x}) = \text{prox}_{\eta\mathcal{R}}(\bar{x} - \eta\nabla\Phi_\gamma(\bar{x}))$. Then, we have

$$\begin{aligned} \|\tilde{x}_\gamma^* - \bar{x}\| &\leq \|\tilde{x}_\gamma^* - \bar{x}_\gamma^*\| + \|\bar{x}_\gamma^* - \bar{x}\| \\ &= \|\text{prox}_{\eta\mathcal{R}}(\bar{x} - \eta\widetilde{\nabla}\Phi_\gamma(\bar{x})) - \text{prox}_{\eta\mathcal{R}}(\bar{x} - \eta\nabla\Phi_\gamma(\bar{x}))\| + \eta\|\mathcal{G}_\eta(\bar{x})\| \\ &\leq \eta\|\widetilde{\nabla}\Phi_\gamma(\bar{x}) - \nabla\Phi_\gamma(\bar{x})\| + \eta\|\mathcal{G}_\eta(\bar{x})\|. \end{aligned} \tag{46}$$

Substituting this estimate into (45), we obtain

$$\|r_x^*\| \leq \left(1 + \eta L_{\Phi_\gamma}\right)\|\mathcal{G}_\eta(\bar{x})\| + (2 + \eta L_{\Phi_\gamma})\|\nabla\Phi_\gamma(\bar{x}) - \widetilde{\nabla}\Phi_\gamma(\bar{x})\|.$$

Combining this inequality and (44), we obtain (42).

Now, since $\tilde{y}_\gamma^* = y_\gamma^*(\tilde{F}(\tilde{x}_\gamma^*))$, by the optimality condition of (9), we have

$$r_y^* := \gamma\nabla b(\tilde{y}_\gamma^*) + K^\top(F(\tilde{x}_\gamma^*) - \tilde{F}(\tilde{x}_\gamma^*)) \in K^\top F(\tilde{x}_\gamma^*) - \partial\psi(\tilde{y}_\gamma^*). \tag{47}$$

Utilizing Lemma A.1(d), we can bound $r_y^*$ defined by (47) as

$$\|r_y^*\| \leq \gamma \|\nabla b(\tilde{y}_\gamma^*)\| + \|K\|\|F(\tilde{x}_\gamma^*) - \tilde{F}(\tilde{x}_\gamma^*)\| \leq \gamma D_\psi + \|K\|\|F(\tilde{x}_\gamma^*) - \tilde{F}(\tilde{x}_\gamma^*)\|.$$

Combining this estimate and (47), we get

$$\mathrm{dist}\left(0, K^\top F(\tilde{x}_\gamma^*) - \partial\psi(\tilde{y}_\gamma^*)\right) \leq \|K\|\|F(\tilde{x}_\gamma^*) - \tilde{F}(\tilde{x}_\gamma^*)\| + \gamma D_\psi. \qquad (48)$$

On the other hand, using the definition of $\widetilde{\nabla}\Phi_\gamma(\cdot)$ from (41), we can show that

$$
\begin{aligned}
\|\widetilde{\nabla}\Phi_\gamma(\bar{x}) - \nabla\Phi_\gamma(\bar{x})\| &= \|\tilde{J}(\bar{x})^\top \nabla\phi_\gamma(\tilde{F}(\bar{x})) - F'(\bar{x})^\top \nabla\phi_\gamma(F(\bar{x}))\| \\
&\leq \|(\tilde{J}(\bar{x}) - F'(\bar{x}))^\top \nabla\phi_\gamma(\tilde{F}(\bar{x}))\| + \|F'(\bar{x})^\top \left(\nabla\phi_\gamma(\tilde{F}(\bar{x})) - \nabla\phi_\gamma(F(\bar{x}))\right)\| \\
&\leq \|(\tilde{J}(\bar{x}) - F'(\bar{x}))^\top \nabla\phi_\gamma(\tilde{F}(\bar{x}))\| + \|\nabla\phi_\gamma(\tilde{F}(\bar{x})) - \nabla\phi_\gamma(F(\bar{x}))\|\|F'(\bar{x})\| \\
&\overset{(i)}{\leq} \|(\tilde{J}(\bar{x}) - F'(\bar{x}))^\top \nabla\phi_\gamma(\tilde{F}(\bar{x}))\| + L_{\phi_\gamma}\|F'(\bar{x})\|\|\tilde{F}(\bar{x}) - F(\bar{x})\| \\
&\overset{(5)}{\leq} \|(\tilde{J}(\bar{x}) - F'(\bar{x}))^\top \nabla\phi_\gamma(\tilde{F}(\bar{x}))\| + L_{\phi_\gamma}M_F\|\tilde{F}(\bar{x}) - F(\bar{x})\|.
\end{aligned}
$$

Here, we have used the $L_{\phi_\gamma}$-smoothness of $\phi_\gamma$ in *(i)*.

Finally, combining the last estimate, (42), and (48), and using the definition of $\mathcal{E}$ from (8), we have

$$
\begin{aligned}
\mathcal{E}(\tilde{x}_\gamma^*, \tilde{y}_\gamma^*) &:= \mathrm{dist}\left(0, \nabla\Phi_\gamma(\tilde{x}_\gamma^*) + \partial\mathcal{R}(\tilde{x}_\gamma^*)\right) + \mathrm{dist}\left(0, K^\top F(\tilde{x}_\gamma^*) - \partial\psi(\tilde{y}_\gamma^*)\right) \\
&\leq \left(1 + \eta L_{\Phi_\gamma}\right)\|\mathcal{G}_\eta(\bar{x})\| + (2 + \eta L_{\Phi_\gamma})\|\nabla\Phi_\gamma(\bar{x}) - \widetilde{\nabla}\Phi_\gamma(\bar{x})\| \\
&\quad + \|K\|\|F(\tilde{x}_\gamma^*) - \tilde{F}(\tilde{x}_\gamma^*)\| + \gamma D_\psi \\
&\leq \left(1 + \eta L_{\Phi_\gamma}\right)\|\mathcal{G}_\eta(\bar{x})\| + \gamma D_\psi + \|K\|\|F(\tilde{x}_\gamma^*) - \tilde{F}(\tilde{x}_\gamma^*)\| \\
&\quad + \left(2 + \eta L_{\Phi_\gamma}\right)\left[\|(\tilde{J}(\bar{x}) - F'(\bar{x}))^\top \nabla\phi_\gamma(\tilde{F}(\bar{x}))\| + L_{\phi_\gamma}M_F\|\tilde{F}(\bar{x}) - F(\bar{x})\|\right],
\end{aligned}
$$

which proves (43). $\qquad\square$

***The proof of Lemma 3.1***. For notational simplicity, we drop the subscript $T$ in this proof. Since $M_{\phi_\gamma} = M_\psi\|K\|$ and $L_{\phi_\gamma} = \frac{\|K\|^2}{\gamma + \mu_\psi}$, using the conditions in Lemma 3.1 and (28), we can derive from (43) after taking the full expectation that

$$
\begin{aligned}
\mathbb{E}\left[\mathcal{E}(\tilde{x}_\gamma^*, \tilde{y}_\gamma^*)\right] &\leq \left(1 + \eta L_{\Phi_\gamma}\right)\mathbb{E}\left[\|\mathcal{G}_\eta(\bar{x})\|\right] + \left(2 + \eta L_{\Phi_\gamma}\right)\mathbb{E}\left[\|(\tilde{J}(\bar{x}) - F'(\bar{x}))^\top \nabla\phi_\gamma(\tilde{F}(\bar{x}))\|\right] \\
&\quad + \|K\|\mathbb{E}\left[\|F(\tilde{x}_\gamma^*) - \tilde{F}(\tilde{x}_\gamma^*)\|\right] + \left(2 + \eta L_{\Phi_\gamma}\right)\frac{\|K\|^2 M_F}{\mu_\psi + \gamma}\mathbb{E}\left[\|\tilde{F}(\bar{x}) - F(\bar{x})\|\right] + \gamma D_\psi.
\end{aligned}
$$

Now, by the Jensen inequality $\mathbb{E}\left[\|\mathcal{G}_\eta(\bar{x})\|\right] \leq \left(\mathbb{E}\left[\|\mathcal{G}_\eta(\bar{x})\|^2\right]\right)^{1/2} \leq \varepsilon$. In addition, by (28), we also have $0 < \gamma \leq c_2\epsilon$, $\mathbb{E}\left[\|(\tilde{J}(\bar{x}) - F'(\bar{x}))^\top \nabla\phi_\gamma(\tilde{F}(\bar{x}))\|\right] \leq \varepsilon$, $\mathbb{E}\left[\|F(\tilde{x}_\gamma^*) - \tilde{F}(\tilde{x}_\gamma^*)\|\right] \leq \varepsilon$, and $\frac{1}{\mu_\psi + \gamma}\mathbb{E}\left[\|\tilde{F}(\bar{x}) - F(\bar{x})\|\right] \leq \varepsilon$. By the update rule of $\eta$ in Theorems 3.1, 3.2, 3.3, and 3.4, we have $\eta L_{\Phi_\gamma} = \frac{2}{3+\theta} \leq \frac{2}{3}$ since $\theta \in (0, 1]$. Substituting these expressions into the last inequality, we finally arrive at

$$\mathbb{E}\left[\mathcal{E}(\tilde{x}_\gamma^*, \tilde{y}_\gamma^*)\right] \leq (1 + \tfrac{2}{3})\varepsilon + c_2 D_\psi\varepsilon + \|K\|\varepsilon + (2 + \tfrac{2}{3})(1 + \|K\|^2 M_F)\varepsilon,$$

which is exactly (30). $\qquad\square$

# B    Convergence analysis of Algorithm 1 in Section 3

This Supp. Doc. provides the full analysis of Algorithm 1, including convergence rates and oracle complexity for both strongly convex and non-strongly convex cases of $\psi$ (or equivalently, the smoothness and the nonsmoothness of $\phi_0$, respectively).

## B.1 Preparing technical results

Let us first recall and prove some technical results to prepare for our convergence analysis.

**Lemma B.1.** *Let $\{x_t\}$ be generated by Algorithm 1, $L_{\Phi_{\gamma_t}}$ be defined by (13), and $B_\psi$ be given in Lemma A.1. Then, under Assumptions 2.1 and 2.2, for any $\eta_t > 0$ and $\theta_t \in [0, 1]$, we have*

$$
\mathbb{E}\big[\Psi_{\gamma_t}(x_{t+1})\big] \leq \mathbb{E}\big[\Psi_{\gamma_{t-1}}(x_t)\big] + \frac{\theta_t\big(1+L_{\Phi_{\gamma_t}}^2\eta_t^2\big)}{2L_{\Phi_{\gamma_t}}}\mathbb{E}\big[\|\nabla\Phi_{\gamma_t}(x_t)-v_t\|^2\big] + (\gamma_{t-1}-\gamma_t)B_\psi
$$
$$
- \frac{L_{\Phi_{\gamma_t}}\eta_t^2\theta_t}{4}\mathbb{E}\big[\|\mathcal{G}_{\eta_t}(x_t)\|^2\big] - \frac{\theta_t}{2}\left(\frac{2}{\eta_t} - L_{\Phi_{\gamma_t}}\theta_t - 2L_{\Phi_{\gamma_t}}\right)\mathbb{E}\big[\|\hat{x}_{t+1}-x_t\|^2\big].
$$
(49)

*Proof.* Following the same line of proof of [35, Lemma 5], we can show that

$$
\mathbb{E}\big[\Psi_{\gamma_t}(x_{t+1})\big] \leq \mathbb{E}\big[\Psi_{\gamma_t}(x_t)\big] + \frac{\theta_t\big(1+L_{\Phi_{\gamma_t}}^2\eta_t^2\big)}{2L_{\Phi_{\gamma_t}}}\mathbb{E}\big[\|\nabla\Phi_{\gamma_t}(x_t)-v_t\|^2\big]
$$
$$
- \frac{L_{\Phi_{\gamma_t}}\eta_t^2\theta_t}{4}\mathbb{E}\big[\|\mathcal{G}_{\eta_t}(x_t)\|^2\big] - \frac{\theta_t}{2}\left(\frac{2}{\eta_t} - L_{\Phi_{\gamma_t}}\theta_t - 2L_{\Phi_{\gamma_t}}\right)\mathbb{E}\big[\|\hat{x}_{t+1}-x_t\|^2\big].
$$

Finally, since $\mathbb{E}\big[\Psi_{\gamma_t}(x_t)\big] \leq \mathbb{E}\big[\Psi_{\gamma_{t-1}}(x_t)\big] + (\gamma_{t-1}-\gamma_t)B_\psi$ due to Lemma A.1(e), substituting this expression into the last inequality, we obtain (49). $\square$

**The Lyapunov function:** To analyze Algorithm 1, we introduce the following Lyapunov function:

$$
V_{\gamma_{t-1}}(x_t) := \mathbb{E}\big[\Psi_{\gamma_{t-1}}(x_t)\big] + \frac{\alpha_t}{2}\mathbb{E}\big[\|\tilde{F}_t - F(x_t)\|^2\big] + \frac{\hat{\alpha}_t}{2}\mathbb{E}\big[\|\tilde{J}_t - F'(x_t)\|^2\big],
\tag{50}
$$

where $\alpha_t > 0$ and $\hat{\alpha}_t > 0$ are given parameters, and the expectation is taken over $\mathcal{F}_{t+1}$. Lemma B.2 provides a key bound to estimate convergence rates and complexity bounds.

**Lemma B.2.** *Let $\{x_t\}$ be generated by Algorithm 1, and $V_{\gamma_t}$ be the Lyapunov function defined by (50). Suppose further that the following conditions hold:*

$$
\begin{cases}
\frac{2}{\eta_t} \geq L_{\Phi_{\gamma_t}}\theta_t + 2L_{\Phi_{\gamma_t}} + \frac{\kappa M_F^2\beta_t^2\theta_t\alpha_{t+1}}{b_1} + \frac{\hat{\kappa}\hat{L}_F^2\hat{\beta}_t^2\theta_t\hat{\alpha}_{t+1}}{\hat{b}_1} \\
2M_F^2 L_{\phi_{\gamma_t}}^2\theta_t\left(\frac{1+L_{\Phi_{\gamma_t}}^2\eta_t^2}{L_{\Phi_{\gamma_t}}}\right) + \alpha_{t+1}\beta_t^2 \leq \alpha_t \quad and \quad 2M_{\phi_{\gamma_t}}^2\theta_t\left(\frac{1+L_{\Phi_{\gamma_t}}^2\eta_t^2}{L_{\Phi_{\gamma_t}}}\right) + \hat{\alpha}_{t+1}\hat{\beta}_t^2 \leq \hat{\alpha}_t.
\end{cases}
\tag{51}
$$

*Then, for all $t \geq 0$, one has*

$$
V_{\gamma_t}(x_{t+1}) \leq V_{\gamma_{t-1}}(x_t) - \frac{L_{\Phi_{\gamma_t}}\eta_t^2\theta_t}{4}\mathbb{E}\big[\|\mathcal{G}_{\eta_t}(x_t)\|^2\big] + \frac{\kappa(1-\beta_t)^2\alpha_{t+1}\sigma_F^2}{b_2} + \frac{\hat{\kappa}(1-\hat{\beta}_t)^2\hat{\alpha}_{t+1}\sigma_J^2}{\hat{b}_2}
$$
$$
+ (\gamma_{t-1}-\gamma_t)B_\psi.
\tag{52}
$$

*Proof.* First of all, by combining (34) and (49), we obtain

$$
\mathbb{E}\big[\Psi_{\gamma_t}(x_{t+1})\big] \leq \mathbb{E}\big[\Psi_{\gamma_{t-1}}(x_t)\big] - \frac{\theta_t}{2}\left(\frac{2}{\eta_t} - L_{\Phi_{\gamma_t}}\theta_t - 2L_{\Phi_{\gamma_t}}\right)\mathbb{E}\big[\|\hat{x}_{t+1}-x_t\|^2\big]
$$
$$
- \frac{L_{\Phi_{\gamma_t}}\eta_t^2\theta_t}{4}\mathbb{E}\big[\|\mathcal{G}_{\eta_t}(x_t)\|^2\big] + (\gamma_{t-1}-\gamma_t)B_\psi
$$
$$
+ \theta_t\left(\frac{1+L_{\Phi_{\gamma_t}}^2\eta_t^2}{L_{\Phi_{\gamma_t}}}\right)\left(M_F^2 L_{\phi_{\gamma_t}}^2\mathbb{E}\big[\|\tilde{F}_t-F(x_t)\|^2\big] + M_{\phi_{\gamma_t}}^2\mathbb{E}\big[\|\tilde{J}_t-F'(x_t)\|^2\big]\right).
\tag{53}
$$

Due to the mini-batch estimators in (15), it is well-known that

$$
\mathbb{E}_{\mathcal{B}_t^2}\big[\|\mathbf{F}(x_t,\zeta_t)-F(x_t)\|^2\big] = \mathbb{E}\big[\big\|\tfrac{1}{b_2}\sum_{\zeta_i\in\mathcal{B}_t^2}\mathbf{F}(x_t,\zeta_i)-F(x_t)\big\|^2\big] \leq \frac{\sigma_F^2}{b_2}
$$
$$
\mathbb{E}_{\hat{\mathcal{B}}_t^2}\big[\|\mathbf{F}'(x_t,\hat{\zeta}_t)-F'(x_t)\|^2\big] = \mathbb{E}\big[\big\|\tfrac{1}{\hat{b}_2}\sum_{\hat{\zeta}_i\in\hat{\mathcal{B}}^2}\mathbf{F}'(x_t,\hat{\zeta}_i)-F'(x_t)\big\|^2\big] \leq \frac{\sigma_J^2}{\hat{b}_2}.
$$

Substituting these bounds and $x_{t+1} - x_t = \theta_t(\hat{x}_{t+1} - x_t)$ into (33) and taking full expectation the resulting inequality over $\mathcal{F}_{t+1}$, we obtain

$$
\mathbb{E}\big[\|\tilde{F}_{t+1}-F(x_{t+1})\|^2\big] \leq \beta_t^2\mathbb{E}\big[\|\tilde{F}_t-F(x_t)\|^2\big] + \frac{\kappa\beta_t^2\theta_t^2 M_F^2}{b_1}\mathbb{E}\big[\|\hat{x}_{t+1}-x_t\|^2\big] + \frac{\kappa(1-\beta_t)^2\sigma_F^2}{b_2}
$$
$$
\mathbb{E}\big[\|\tilde{J}_{t+1}-F'(x_{t+1})\|^2\big] \leq \hat{\beta}_t^2\mathbb{E}\big[\|\tilde{J}_t-F'(x_t)\|^2\big] + \frac{\hat{\kappa}\hat{\beta}_t^2\theta_t^2 \hat{L}_F^2}{\hat{b}_1}\mathbb{E}\big[\|\hat{x}_{t+1}-x_t\|^2\big] + \frac{\hat{\kappa}(1-\hat{\beta}_t)^2\sigma_J^2}{\hat{b}_2}.
$$

Multiplying these inequalities by $\alpha_{t+1} > 0$ and $\hat{\alpha}_{t+1} > 0$, respectively, and adding the results to (53), we can further derive

$$V_{\gamma_t}(x_{t+1}) \overset{(50)}{:=} \mathbb{E}\big[\Psi_{\gamma_t}(x_{t+1})\big] + \tfrac{\alpha_{t+1}}{2}\mathbb{E}\big[\|\tilde{F}_{t+1} - F(x_{t+1})\|^2\big] + \tfrac{\hat{\alpha}_{t+1}}{2}\mathbb{E}\big[\|\tilde{J}_{t+1} - F'(x_{t+1})\|^2\big]$$

$$\leq \mathbb{E}\big[\Psi_{\gamma_{t-1}}(x_t)\big] + \left[M_F^2 L_{\phi_{\gamma_t}}^2 \theta_t \Big(\tfrac{1+L_{\Phi_{\gamma_t}}^2 \eta_t^2}{L_{\Phi_{\gamma_t}}}\Big) + \tfrac{\alpha_{t+1}\beta_t^2}{2}\right]\mathbb{E}\big[\|\tilde{F}_t - F(x_t)\|^2\big]$$

$$+ \left[M_{\phi_{\gamma_t}}^2 \theta_t \Big(\tfrac{1+L_{\Phi_{\gamma_t}}^2 \eta_t^2}{L_{\Phi_{\gamma_t}}}\Big) + \tfrac{\hat{\alpha}_{t+1}\hat{\beta}_t^2}{2}\right]\mathbb{E}\big[\|\tilde{J}_t - F'(x_t)\|^2\big] - \tfrac{L_{\Phi_{\gamma_t}}\eta_t^2 \theta_t}{4}\mathbb{E}\big[\|\mathcal{G}_{\eta_t}(x_t)\|^2\big]$$

$$- \tfrac{\theta_t}{2}\Big(\tfrac{2}{\eta_t} - L_{\Phi_{\gamma_t}}\theta_t - 2L_{\Phi_{\gamma_t}} - \tfrac{\kappa M_F^2 \beta_t^2 \theta_t \alpha_{t+1}}{b_1} - \tfrac{\hat{\kappa}L_F^2 \hat{\beta}_t^2 \theta_t \hat{\alpha}_{t+1}}{\hat{b}_1}\Big)\mathbb{E}\big[\|\hat{x}_{t+1} - x_t\|^2\big]$$

$$+ \tfrac{\kappa(1-\beta_t)^2 \alpha_{t+1}\sigma_F^2}{b_2} + \tfrac{\hat{\kappa}(1-\hat{\beta}_t)^2 \hat{\alpha}_{t+1}\sigma_J^2}{\hat{b}_2} + (\gamma_{t-1} - \gamma_t)B_\psi.$$

Let us choose $\alpha_t > 0$ and $\hat{\alpha}_t > 0$ and impose three conditions as in (51), i.e.:

$$\begin{cases} \tfrac{2}{\eta_t} \geq L_{\Phi_{\gamma_t}}\theta_t + 2L_{\Phi_{\gamma_t}} + \tfrac{\kappa M_F^2 \beta_t^2 \theta_t \alpha_{t+1}}{b_1} + \tfrac{\hat{\kappa}L_F^2 \hat{\beta}_t^2 \theta_t \hat{\alpha}_{t+1}}{\hat{b}_1}, \\[2mm] 2M_F^2 L_{\phi_{\gamma_t}}^2 \theta_t \Big(\tfrac{1+L_{\Phi_{\gamma_t}}^2 \eta_t^2}{L_{\Phi_{\gamma_t}}}\Big) + \alpha_{t+1}\beta_t^2 \leq \alpha_t, \quad \text{and} \quad 2M_{\phi_{\gamma_t}}^2 \theta_t \Big(\tfrac{1+L_{\Phi_{\gamma_t}}^2 \eta_t^2}{L_{\Phi_{\gamma_t}}}\Big) + \hat{\alpha}_{t+1}\hat{\beta}_t^2 \leq \hat{\alpha}_t. \end{cases}$$

Then, by using (50), the last inequality can be further upper bounded as

$$V_{\gamma_t}(x_{t+1}) \leq V_{\gamma_{t-1}}(x_t) - \tfrac{L_{\Phi_{\gamma_t}}\eta_t^2 \theta_t}{4}\mathbb{E}\big[\|\mathcal{G}_{\eta_t}(x_t)\|^2\big] + \tfrac{\kappa(1-\beta_t)^2 \alpha_{t+1}\sigma_F^2}{b_2}$$

$$+ \tfrac{\hat{\kappa}(1-\hat{\beta}_t)^2 \hat{\alpha}_{t+1}\sigma_J^2}{\hat{b}_2} + (\gamma_{t-1} - \gamma_t)B_\psi,$$

which proves (52). □

## B.2  A general key bound for Algorithm 1

Now, we are ready to prove one key result, Theorem B.1, for oracle complexity analysis of Algorithm 1. To simplify our expressions, let us introduce the following notations in advance:

$$\begin{cases} \omega_t \;:=\; \tfrac{\theta_t}{L_{\Phi_{\gamma_t}}} \quad \text{and} \quad \Sigma_T := \sum_{t=0}^{T}\omega_t, \\[3mm] \Theta_t \;:=\; \tfrac{M_F^2 L_{\phi_{\gamma_t}}^2 \sqrt{26 b_1 \hat{b}_1}}{3\big(\kappa M_F^4 L_{\phi_{\gamma_t}}^2 \hat{b}_1 + \hat{\kappa}M_{\phi_{\gamma_t}}^2 L_F^2 b_1\big)^{1/2}}, \\[3mm] \Pi_0 \;:=\; \tfrac{\sqrt{26 b_1 \hat{b}_1}}{3\big(\hat{b}_1 \kappa M_F^4 L_{\phi_{\gamma_0}}^2 + b_1 \hat{\kappa}L_F^2 M_{\phi_{\gamma_0}}^2\big)^{1/2}}\left(\tfrac{\kappa M_F^2 L_{\phi_{\gamma_0}}^2 \sigma_F^2}{b_0} + \tfrac{\hat{\kappa}M_{\phi_{\gamma_0}}^2 \sigma_J^2}{\hat{b}_0}\right), \\[3mm] \Gamma_t \;:=\; \tfrac{\sqrt{26 b_1 \hat{b}_1}}{3\big(\hat{b}_1 \kappa M_F^4 L_{\phi_{\gamma_t}}^2 + b_1 \hat{\kappa}L_F^2 M_{\phi_{\gamma_t}}^2\big)^{1/2}}\left(\tfrac{\kappa M_F^2 L_{\phi_{\gamma_t}}^2 \sigma_F^2}{b_2} + \tfrac{\hat{\kappa}M_{\phi_{\gamma_t}}^2 \sigma_J^2}{\hat{b}_2}\right). \end{cases} \quad (54)$$

**Theorem B.1.** *Suppose that Assumptions 2.1 and 2.2 hold, and $\omega_t$, $\Sigma_T$, $\Theta_t$, $\Pi_0$, and $\Gamma_t$ are defined by (54). Let $\{x_t\}_{t=0}^{T}$ be generated by Algorithm 1 using the following step-sizes:*

$$\theta_t := \frac{3L_{\Phi_{\gamma_t}}\big[b_1 \hat{b}_1(1-\beta_t)\big]^{1/2}}{\sqrt{26}(\kappa M_F^4 L_{\phi_{\gamma_t}}^2 \hat{b}_1 + \hat{\kappa}M_{\phi_{\gamma_t}}^2 L_F^2 b_1)^{1/2}} \quad \text{and} \quad \eta_t := \frac{2}{L_{\Phi_{\gamma_t}}(3+\theta_t)}, \quad (55)$$

*where $\beta_t, \hat{\beta}_t \in (0,1]$ are chosen such that $\beta_t = \hat{\beta}_t$, $0 \leq \gamma_{t+1} \leq \gamma_t$, and*

$$\begin{cases} \tfrac{\beta_t^2(1-\beta_t)}{\Theta_t^2} \leq \tfrac{1-\beta_{t+1}}{\Theta_{t+1}^2} \leq \tfrac{1-\beta_t}{\Theta_t^2}, \\[3mm] \beta_t > \max\left\{0, 1 - \tfrac{26}{9L_{\Phi_{\gamma_t}}^2}\Big(\tfrac{\kappa M_F^4 L_{\phi_{\gamma_t}}^2}{b_1} + \tfrac{\hat{\kappa}L_F^2 M_{\phi_{\gamma_t}}^2}{\hat{b}_1}\Big)\right\}. \end{cases} \quad (56)$$

*Let $\bar{x}_T$ be randomly chosen between $\{x_0, \cdots, x_T\}$ such that $\mathbf{Prob}\,(\bar{x}_T = x_t) = \tfrac{\omega_t}{\Sigma_T}$, and $\bar{\eta}_T$ be corresponding to $\eta_t$ of $\bar{x}_T$. Then, the following estimate holds:*

$$\mathbb{E}\big[\|\mathcal{G}_{\bar{\eta}_T}(\bar{x}_T)\|^2\big] \leq \frac{16}{\Sigma_T}\Big(\mathbb{E}\big[\Psi_0(x_0) - \Psi_0^\star\big] + \gamma_T B_\psi\Big) + \frac{8\Pi_0}{\Sigma_T \sqrt{1-\beta_0}} + \frac{16}{\Sigma_T}\sum_{t=0}^{T}\frac{\Gamma_{t+1}(1-\beta_t)^2}{\sqrt{1-\beta_{t+1}}}. \quad (57)$$

***The proof of Theorem B.1***. First, the conditions in (51) can be simplified as follows:

$$
\begin{cases}
L_{\Phi_{\gamma_t}}\theta_t + 2L_{\Phi_{\gamma_t}} + \big(\frac{\kappa M_F^2 \beta_t^2 \alpha_{t+1}}{b_1} + \frac{\hat{\kappa}L_F^2 \hat{\beta}_t^{\,2}\hat{\alpha}_{t+1}}{\hat{b}_1}\big)\theta_t & \leq & \frac{2}{\eta_t}, \\
2M_F^2 L_{\phi_{\gamma_t}}^2 (1 + L_{\Phi_{\gamma_t}}^2 \eta_t^2)\theta_t & \leq & L_{\Phi_{\gamma_t}}(\alpha_t - \beta_t^2 \alpha_{t+1}), \\
2M_{\phi_{\gamma_t}}^2 (1 + L_{\Phi_{\gamma_t}}^2 \eta_t^2)\theta_t & \leq & L_{\Phi_{\gamma_t}}(\hat{\alpha}_t - \hat{\beta}_t^2 \hat{\alpha}_{t+1}).
\end{cases}
\tag{58}
$$

Let us update $\eta_t := \frac{2}{(3+\theta_t)L_{\Phi_{\gamma_t}}}$ as (55). Since $\theta_t \in (0,1]$, we have

$$
\frac{1}{2L_{\Phi_{\gamma_t}}} \leq \eta_t < \frac{2}{3L_{\Phi_{\gamma_t}}} \quad \text{and} \quad 1 \leq 1 + L_{\Phi_{\gamma_t}}^2 \eta_t^2 < \frac{13}{9}.
$$

Next, let us choose $\gamma_t$, $\beta_t$, $\hat{\beta}_t$, $\alpha_t$, and $\hat{\alpha}_t$ such that

$$
\hat{\beta}_t = \beta_t \in (0,1], \quad \hat{\alpha}_t = \frac{M_{\phi_{\gamma_t}}^2}{M_F^2 L_{\phi_{\gamma_t}}^2}\alpha_t, \quad \frac{M_{\phi_{\gamma_{t+1}}}}{L_{\phi_{\gamma_{t+1}}}} \leq \frac{M_{\phi_{\gamma_t}}}{L_{\phi_{\gamma_t}}}, \quad \text{and} \quad 0 < \alpha_t \leq \alpha_{t+1} \leq \frac{\alpha_t}{\beta_t}.
\tag{59}
$$

Then, we have

$$
\alpha_t - \alpha_{t+1}\beta_t^2 \geq \alpha_t(1 - \beta_t) > 0,
$$

and

$$
\hat{\alpha}_t - \hat{\beta}_t^2 \hat{\alpha}_{t+1} = \frac{M_{\phi_{\gamma_t}}^2}{M_F^2 L_{\phi_{\gamma_t}}^2}\alpha_t - \beta_t^2 \frac{M_{\phi_{\gamma_{t+1}}}^2}{M_F^2 L_{\phi_{\gamma_{t+1}}}^2}\alpha_{t+1} \geq \frac{M_{\phi_{\gamma_t}}^2}{M_F^2 L_{\phi_{\gamma_t}}^2}(\alpha_t - \beta_t^2 \alpha_{t+1})
$$

$$
\geq \frac{M_{\phi_{\gamma_t}}^2}{M_F^2 L_{\phi_{\gamma_t}}^2}(1 - \beta_t)\alpha_t = (1 - \beta_t)\hat{\alpha}_t > 0.
$$

By using the last two inequalities, we can show that the conditions in (58) hold, if we have

$$
0 < \theta_t \leq \frac{9L_{\Phi_{\gamma_t}}\alpha_t(1-\beta_t)}{26M_F^2 L_{\phi_{\gamma_t}}^2}, \qquad 0 < \theta_t \leq \frac{9L_{\Phi_{\gamma_t}}\hat{\alpha}_t(1-\beta_t)}{26M_{\phi_{\gamma_t}}^2},
$$

$$
\text{and} \qquad 0 < \theta_t \leq L_{\Phi_{\gamma_t}}\left(\frac{\kappa M_F^2 \alpha_t}{b_1} + \frac{\hat{\kappa}L_F^2 \hat{\alpha}_t}{\hat{b}_1}\right)^{-1}.
\tag{60}
$$

Therefore, the three conditions in (60) hold if we choose

$$
\frac{\alpha_t(1-\beta_t)}{M_F^2 L_{\phi_{\gamma_t}}^2} = \frac{\hat{\alpha}_t(1-\beta_t)}{M_{\phi_{\gamma_t}}^2} \quad \text{and} \quad \left(\frac{\kappa M_F^2}{b_1} + \frac{\hat{\kappa}L_F^2 M_{\phi_{\gamma_t}}^2}{M_F^2 L_{\phi_{\gamma_t}}^2 \hat{b}_1}\right)\alpha_t = \frac{26M_F^2 L_{\phi_{\gamma_t}}^2}{9\alpha_t(1-\beta_t)}.
$$

These conditions show that we can choose

$$
\alpha_t := \frac{\Theta_t}{\sqrt{1-\beta_t}} \quad \text{and} \quad \hat{\alpha}_t := \frac{M_{\phi_{\gamma_t}}^2 \Theta_t}{M_F^2 L_{\phi_{\gamma_t}}^2 \sqrt{1-\beta_t}}, \quad \text{where} \quad \Theta_t := \frac{M_F^2 L_{\phi_{\gamma_t}}^2 \sqrt{26b_1 \hat{b}_1}}{3\left(\kappa M_F^4 L_{\phi_{\gamma_t}}^2 \hat{b}_1 + \hat{\kappa}M_{\phi_{\gamma_t}}^2 L_F^2 b_1\right)^{1/2}}.
$$

Clearly, this $\Theta_t$ is exactly given by (54). With this choice of $\alpha_t$ and $\hat{\alpha}_t$, we obtain

$$
0 < \theta_t \leq \bar{\theta}_t := \frac{9L_{\Phi_{\gamma_t}}\Theta_t \sqrt{(1-\beta_t)}}{26M_F^2 L_{\phi_{\gamma_t}}^2} = \frac{3L_{\Phi_{\gamma_t}}\sqrt{b_1 \hat{b}_1(1-\beta_t)}}{\sqrt{26}(\kappa M_F^4 L_{\phi_{\gamma_t}}^2 \hat{b}_1 + \hat{\kappa}M_{\phi_{\gamma_t}}^2 L_F^2 b_1)^{1/2}}.
$$

We then choose $\theta_t := \bar{\theta}_t$ at the upper bound as in (55).

Now, to guarantee that $0 < \bar{\theta}_t \leq 1$, we impose the following condition as in (56), i.e.:

$$
\beta_t > \max\left\{0, 1 - \frac{26}{9L_{\Phi_{\gamma_t}}^2}\left(\frac{\kappa M_F^4 L_{\phi_{\gamma_t}}^2}{b_1} + \frac{\hat{\kappa}L_F^2 M_{\phi_{\gamma_t}}^2}{\hat{b}_1}\right)\right\}.
$$

Due to the choice of $\alpha_t$, the condition $\alpha_t \leq \alpha_{t+1} \leq \frac{\alpha_t}{\beta_t}$ in (59) is equivalent to

$$
\frac{\beta_t^2(1-\beta_t)}{\Theta_t^2} \leq \frac{1-\beta_{t+1}}{\Theta_{t+1}^2} \leq \frac{1-\beta_t}{\Theta_t^2},
$$

which is the first condition of (56). Moreover, since $M_{\phi_{\gamma_t}} = M_\psi \|K\|$ and $L_{\phi_{\gamma_t}} = \frac{\|K\|^2}{\mu_\psi + \gamma_t}$ due to Lemma A.1, the third condition of (59) reduces to $\gamma_{t+1} \leq \gamma_t$, which is one of the conditions in Theorem B.1.

Next, under the choice of $\alpha_t$ and $\hat{\alpha}_t$, and $\eta_t \geq \frac{1}{2L_{\Phi_{\gamma_t}}}$, (52) implies

$$\frac{\theta_t}{16L_{\Phi_{\gamma_t}}}\mathbb{E}\big[\|\mathcal{G}_{\eta_t}(x_t)\|^2\big] \leq V_{\gamma_{t-1}}(x_t) - V_{\gamma_t}(x_{t+1}) + (\gamma_{t-1} - \gamma_t)B_\psi$$
$$+ \frac{\sqrt{26b_1\hat{b}_1}}{3\big(\hat{b}_1\kappa M_F^4 L_{\phi_{\gamma_{t+1}}}^2 + b_1\hat{\kappa}L_F^2 M_{\phi_{\gamma_{t+1}}}^2\big)^{1/2}} \left( \frac{\kappa M_F^2 L_{\phi_{\gamma_{t+1}}}^2 \sigma_F^2}{b_2} + \frac{\hat{\kappa}M_{\phi_{\gamma_{t+1}}}^2 \sigma_J^2}{\hat{b}_2} \right) \frac{(1-\beta_t)^2}{(1-\beta_{t+1})^{1/2}}. \quad (61)$$

Note that since $\Psi_{\gamma_0}(x_0) \leq \Psi_0(x_0)$ due to Lemma A.1, and $\gamma_{-1} = \gamma_0$ by convention, we have

$$V_{\gamma_0}(x_0) = \mathbb{E}\big[\Psi_{\gamma_0}(x_0)\big] + \frac{\alpha_0}{2}\mathbb{E}\big[\|\tilde{F}_0 - F(x_0)\|^2\big] + \frac{\hat{\alpha}_0}{2}\mathbb{E}\big[\|\tilde{J}_0 - F'(x_0)\|^2\big]$$
$$\leq \mathbb{E}\big[\Psi_0(x_0)\big] + \frac{\sqrt{26b_1\hat{b}_1}}{6\big(\hat{b}_1\kappa M_F^4 L_{\phi_{\gamma_0}}^2 + b_1\hat{\kappa}L_F^2 M_{\phi_{\gamma_0}}^2\big)^{1/2}} \left( \frac{\kappa M_F^2 L_{\phi_{\gamma_0}}^2 \sigma_F^2}{b_0} + \frac{\hat{\kappa}M_{\phi_{\gamma_0}}^2 \sigma_J^2}{\hat{b}_0} \right) \frac{1}{(1-\beta_0)^{1/2}}. \quad (62)$$

Moreover, by Lemma A.1(d), we have

$$V_{\gamma_T}(x_{T+1}) \geq \mathbb{E}\big[\Psi_{\gamma_T}(x_{T+1})\big] \geq \mathbb{E}\big[\Psi_0(x_{T+1})\big] - \gamma_T B_\psi \geq \Psi_0^\star - \gamma_T B_\psi. \quad (63)$$

Let us define $\Gamma_t$ and $\Pi_0$ as (54), i.e.:

$$\begin{cases} \Gamma_t & := \dfrac{\sqrt{26b_1\hat{b}_1}}{3\big(\hat{b}_1\kappa M_F^4 L_{\phi_{\gamma_t}}^2 + b_1\hat{\kappa}L_F^2 M_{\phi_{\gamma_t}}^2\big)^{1/2}} \left( \dfrac{\kappa M_F^2 L_{\phi_{\gamma_t}}^2 \sigma_F^2}{b_2} + \dfrac{\hat{\kappa}M_{\phi_{\gamma_t}}^2 \sigma_J^2}{\hat{b}_2} \right), \\[3ex] \Pi_0 & := \dfrac{\sqrt{26b_1\hat{b}_1}}{3\big(\hat{b}_1\kappa M_F^4 L_{\phi_{\gamma_0}}^2 + b_1\hat{\kappa}L_F^2 M_{\phi_{\gamma_0}}^2\big)^{1/2}} \left( \dfrac{\kappa M_F^2 L_{\phi_{\gamma_0}}^2 \sigma_F^2}{b_0} + \dfrac{\hat{\kappa}M_{\phi_{\gamma_0}}^2 \sigma_J^2}{\hat{b}_0} \right). \end{cases}$$

Then, summing up (61) from $t := 0$ to $t := T$, and using these expressions, (62), and (63), we get

$$\sum_{t=0}^{T} \frac{\theta_t}{16L_{\Phi_{\gamma_t}}}\mathbb{E}\big[\|\mathcal{G}_{\eta_t}(x_t)\|^2\big] \leq \mathbb{E}\big[\Psi_0(x_0) - \Psi_0^\star\big] + \gamma_T B_\psi + \sum_{t=0}^{T} \frac{\Gamma_{t+1}(1-\beta_t)^2}{(1-\beta_{t+1})^{1/2}} + \frac{\Pi_0}{2(1-\beta_0)^{1/2}}.$$

Dividing this inequality by $\frac{\Sigma_T}{16}$, where $\Sigma_T := \sum_{t=0}^{T} \omega_t \equiv \sum_{t=0}^{T} \frac{\theta_t}{L_{\Phi_{\gamma_t}}}$, we obtain

$$\frac{1}{\Sigma_T}\sum_{t=0}^{T} \omega_t \mathbb{E}\big[\|\mathcal{G}_{\eta_t}(x_t)\|^2\big] \leq \frac{16}{\Sigma_T}\left( \mathbb{E}\big[\Psi_0(x_0) - \Psi_0^\star\big] + \gamma_T B_\psi \right) + \frac{8\Pi_0}{\Sigma_T(1-\beta_0)^{1/2}}$$
$$+ \frac{16}{\Sigma_T}\sum_{t=0}^{T} \frac{\Gamma_{t+1}(1-\beta_t)^2}{(1-\beta_{t+1})^{1/2}}.$$

Finally, due to the choice of $\bar{x}_T$ and $\bar{\eta}_T$, we have $\frac{1}{\Sigma_T}\sum_{t=0}^{T} \omega_t \mathbb{E}\big[\|\mathcal{G}_{\eta_t}(x_t)\|^2\big] = \mathbb{E}\big[\|\mathcal{G}_{\bar{\eta}_T}(\bar{x}_T)\|^2\big]$. This relation together with the above estimate prove (57). $\qquad \square$

## B.3  The proof of Theorem 3.1: The smooth case with constant step-size

Now, we prove our first main result in the main text.

***The proof of Theorem 3.1 in the main text.*** First, since $\mu_\psi = 1 > 0$, we can set $\gamma_t := 0$ for all $t \geq 0$. That means, we do not need to smooth $\phi_0$ in (2). Hence, from (54), $\Theta_t = \Theta_0 = \frac{M_F^2 L_{\phi_0}\sqrt{26b_1\hat{b}_1}}{3\big(\kappa M_F^4 L_{\phi_0}^2 \hat{b}_1 + \hat{\kappa}M_{\phi_0}^2 L_F^2 b_1\big)^{1/2}}$ and $\frac{\omega_t}{\Sigma_T} = \frac{\theta_t}{\sum_{t=0}^{T}\theta_t}$, where $L_{\Phi_0}$ is defined by (19).

Next, given a batch size $b > 0$, let us choose the mini-batch sizes $b_0 := c_0\hat{b}_0 > 0$, $\hat{b}_1 := \hat{b}_2 := b > 0$, and $b_1 = b_2 := c_0 b$ for some $c_0 > 0$. We also choose a constant step-size $\theta_t := \theta \in (0, 1]$ and a constant weight $\beta_t := \beta \in (0, 1]$ for all $t \geq 0$. We also recall $P$, $Q$, and $L_{\Phi_0}$ defined by (19).

With this configuration, the first condition of (56) and $0 \leq \gamma_{t+1} \leq \gamma_t$ are automatically satisfied, while the second one becomes

$$\beta > \max\left\{ 0, 1 - \frac{26}{9c_0 L_{\Phi_0}^2 b}\big(\kappa M_F^4 \|K\|^4 + c_0\hat{\kappa}\|K\|^2 L_F^2 M_\psi^2\big) \right\} = \max\left\{ 0, 1 - \frac{P^2}{L_{\Phi_0}^2 b} \right\}. \quad (64)$$

Moreover, we also obtain from (54), (55), and (19) that

$$
\begin{cases}
\theta_t &=& \theta = \dfrac{3L_{\Phi_0}\sqrt{c_0 b(1-\beta)}}{\sqrt{26}(\kappa M_F^4 \|K\|^4 + c_0\hat\kappa\|K\|^2 M_\psi^2 L_F^2)^{1/2}} &\overset{(19)}{=}& \dfrac{L_{\Phi_0}[b(1-\beta)]^{1/2}}{P}, \\[3mm]
\Gamma_t &=& \Gamma = \dfrac{\sqrt{26}\big(\kappa M_F^4\|K\|^4\sigma_F^2 + c_0\hat\kappa\|K\|^2 M_\psi^2\sigma_J^2\big)}{3\sqrt{c_0 b}\big(\kappa M_F^4\|K\|^4 + c_0\hat\kappa\|K\|^2 L_F^2 M_\psi^2\big)^{1/2}} &\overset{(19)}{=}& \dfrac{Q}{P\sqrt b}, \\[3mm]
\Pi_0 &=& \dfrac{\sqrt{26 b}\big(\kappa M_F^2\|K\|^4\sigma_F^2 + c_0\hat\kappa\|K\|^2 M_\psi^2\sigma_J^2\big)}{3\sqrt{c_0}\hat b_0\big(\kappa M_F^4\|K\|^4 + c_0\hat\kappa\|K\|^2 L_F^2 M_\psi^2\big)^{1/2}} &\overset{(19)}{=}& \dfrac{Q\sqrt b}{P\hat b_0}, \\[3mm]
\Sigma_T &=& \sum_{t=0}^{T}\dfrac{\theta}{L_{\Phi_0}} = \dfrac{\theta(T+1)}{L_{\Phi_0}} &=& \dfrac{(T+1)[b(1-\beta)]^{1/2}}{P}.
\end{cases}
$$

Furthermore, with these expressions of $\Gamma_t$, $\Pi_0$, and $\Sigma_T$, (57) reduces to

$$
\mathbb{E}\big[\|\mathcal{G}_\eta(\bar x_T)\|^2\big] \leq \frac{16P}{(T+1)[b(1-\beta)]^{1/2}}\mathbb{E}\big[\Psi_0(x_0) - \Psi_0^\star\big] + \frac{8Q}{\hat b_0(T+1)(1-\beta)} + \frac{16Q(1-\beta)}{b}.
$$

Trading-off the term $\frac{1}{\hat b_0(1-\beta)(T+1)} + \frac{2(1-\beta)}{b}$ over $\beta \in (0,1]$, we obtain $\beta := 1 - \frac{b^{1/2}}{[\hat b_0(T+1)]^{1/2}}$, which has shown in (20). In this case, $\theta_t = \theta = \frac{L_{\Phi_0}[b(1-\beta)]^{1/2}}{P} = \frac{L_{\Phi_0}b^{3/4}}{P[\hat b_0(T+1)]^{1/4}}$ as shown in (20).

Now, let us choose $\hat b_0 := c_1^2[b(T+1)]^{1/3}$ for some $c_1 > 0$. Then, the last inequality leads to

$$
\mathbb{E}\big[\|\mathcal{G}_\eta(\bar x_T)\|^2\big] \leq \frac{16P\sqrt{c_1}}{[b(T+1)]^{2/3}}\big[\Psi_0(x_0) - \Psi_0^\star\big] + \frac{24Q}{2c_1[b(T+1)]^{2/3}}.
$$

Hence, if we define $\Delta_0$ as in (21), i.e.:

$$
\Delta_0 := 16P\sqrt{c_1}\big[\Psi_0(x_0) - \Psi_0^\star\big] + \frac{24Q}{c_1},
$$

then we obtain from the last inequality that (21) holds, i.e.:

$$
\mathbb{E}\big[\|\mathcal{G}_\eta(\bar x_T)\|^2\big] \leq \frac{\Delta_0}{[b(T+1)]^{2/3}}.
$$

Consequently, for a given tolerance $\varepsilon > 0$, to obtain $\mathbb{E}\big[\|\mathcal{G}_\eta(\bar x_T)\|^2\big] \leq \varepsilon^2$, we need at most $T := \big\lfloor \frac{\Delta_0^{3/2}}{b\varepsilon^3}\big\rfloor$ iterations. In this case, the total number of function evaluations $\mathbf{F}(x_t, \xi)$ is at most

$$
\mathcal{T}_F := b_0 + (T+1)(2b_1 + b_2) = c_0 c_1^2[b(T+1)]^{1/3} + 3c_0(T+1)b = \frac{c_0 c_1^2 \Delta_0^{1/2}}{\varepsilon} + \frac{3c_0\Delta_0^{3/2}}{\varepsilon^3}.
$$

Alternatively, the total number of Jacobian evaluations $\mathbf{F}'(x_t, \xi)$ is at most

$$
\mathcal{T}_J := \hat b_0 + (T+1)(2\hat b_1 + \hat b_2) = c_1^2[b(T+1)]^{1/3} + 3(T+1)b = \frac{c_1^2\Delta_0^{1/2}}{\varepsilon} + \frac{3\Delta_0^{3/2}}{\varepsilon^3}.
$$

Finally, since $\beta := 1 - \frac{b^{1/2}}{[\hat b_0(T+1)]^{1/2}}$, the condition (64) leads to $\frac{b^{1/2}}{[\hat b_0(T+1)]^{1/2}} < \frac{P^2}{L_{\Phi_0}^2 b}$, which is equivalent to $\frac{\hat b_0(T+1)}{b^3} > \frac{L_{\Phi_0}^4}{P^4}$ as shown in Theorem 3.1. $\qquad\square$

## B.4 The proof of Theorem 3.2: The smooth case with diminishing step-size

***The proof of Theorem 3.2 in the main text.*** Similar to the proof of Theorem 3.1, with $\mu_\psi = 1 > 0$, we set $\gamma_t = 0$. Hence, we obtain $\Theta_t = \Theta_0 = \dfrac{M_F^2 L_{\phi_0}\sqrt{26 b_1\hat b_1}}{3\big(\kappa M_F^4 L_{\phi_0}^2\hat b_1 + \hat\kappa M_{\phi_0}^2 L_F^2 b_1\big)^{1/2}}$ and $\dfrac{\omega_t}{\Sigma_T} = \dfrac{\theta_t}{\sum_{t=0}^{T}\theta_t}$.

Next, given a mini-batch size $b > 0$, let us choose the mini-batch sizes $b_0 := c_0\hat b_0$, $\hat b_1 = \hat b_2 := b$, and $b_1 = b_2 := c_0 b > 0$ for some $c_0 > 0$. With these choices, the condition (56) becomes

$$
\beta_t^2(1-\beta_t) \leq 1-\beta_{t+1} \leq 1-\beta_t \text{ and } \beta_t > \max\left\{0, 1 - \frac{26}{9c_0 L_{\Phi_0}^2 b}\big(c_0\kappa M_F^4 L_{\phi_0}^2 + \hat\kappa L_F^2 M_{\phi_0}^2\big)\right\}. \tag{65}
$$

Moreover, from (54) and (55), we have

$$
\begin{cases}
\theta_t &= \dfrac{3L_{\Phi_0}\sqrt{c_0 b(1-\beta_t)}}{\sqrt{26}(\kappa M_F^4\|K\|^4 + c_0\hat{\kappa}\|K\|^2 M_\psi^2 L_F^2)^{1/2}} & \overset{(19)}{=} \ \dfrac{L_{\Phi_0}[b(1-\beta_t)]^{1/2}}{P}, \\[4mm]
\Gamma_t &= \Gamma = \dfrac{\sqrt{26}\big(\kappa M_F^2\|K\|^4\sigma_F^2 + c_0\hat{\kappa}\|K\|^2 M_\psi^2\sigma_J^2\big)}{3\sqrt{c_0 b}\big(\kappa M_F^4\|K\|^4 + c_0\hat{\kappa}\|K\|^2 L_F^2 M_\psi^2\big)^{1/2}} & \overset{(19)}{=} \ \dfrac{Q}{P\sqrt{b}}, \\[4mm]
\Pi_0 &= \dfrac{\sqrt{26b}\big(\kappa M_F^2\|K\|^4\sigma_F^2 + c_0\hat{\kappa}\|K\|^2 M_\psi^2\sigma_J^2\big)}{3\sqrt{c_0}\hat{b}_0\big(\kappa M_F^4\|K\|^4 + c_0\hat{\kappa}\|K\|^2 L_F^2 M_\psi^2\big)^{1/2}} & \overset{(19)}{=} \ \dfrac{Q\sqrt{b}}{P\hat{b}_0}, \\[4mm]
\Sigma_T &= \sum_{t=0}^{T}\omega_t = \sum_{t=0}^{T}\dfrac{\theta_t}{L_{\Phi_0}} & = \ \dfrac{\sqrt{b}}{P}\sum_{t=0}^{T}\sqrt{1-\beta_t}.
\end{cases}
$$

Furthermore, with these expressions of $\Gamma_t$, $\Pi_0$, and $\Sigma_T$, (57) reduces to

$$
\frac{1}{\sum_{t=0}^{T}\theta_t}\sum_{t=0}^{T}\theta_t\mathbb{E}\big[\|\mathcal{G}_{\eta_t}(x_t)\|^2\big] \le \frac{16P}{\sqrt{b}\sum_{t=0}^{T}\sqrt{1-\beta_t}}\big[\Psi_0(x_0) - \Psi_0^\star\big] + \frac{8Q}{\hat{b}_0\sqrt{1-\beta_0}\sum_{t=0}^{T}\sqrt{1-\beta_t}}
$$
$$
+ \frac{16Q}{b\sum_{t=0}^{T}\sqrt{1-\beta_t}}\sum_{t=0}^{T}\frac{(1-\beta_t)^2}{(1-\beta_{t+1})^{1/2}}. \tag{66}
$$

Let us choose $\beta_t := 1 - \frac{1}{(t+2)^{2/3}} \in (0,1)$ as in (22). Then, it is easy to check that $\beta_t^2(1-\beta_t) \le 1 - \beta_{t+1} \le 1 - \beta_t$ after a few elementary calculations.

Moreover, we have $\theta_t := \frac{L_{\Phi_0}\sqrt{b}}{P(t+2)^{1/3}}$ as (22). In addition, one can easily show that

$$
\begin{cases}
\sum_{t=0}^{T}\sqrt{1-\beta_t} = \sum_{t=0}^{T}\frac{1}{(t+2)^{1/3}} \ge \int_2^{T+3}\frac{ds}{s^{1/3}} = \frac{3}{2}[(T+3)^{2/3} - 2^{2/3}], \\[3mm]
\sum_{t=0}^{T}\frac{(1-\beta_t)^2}{\sqrt{1-\beta_{t+1}}} = \sum_{t=0}^{T}\frac{(t+3)^{1/3}}{(t+2)^{4/3}} \le \sum_{t=0}^{T}\frac{1}{(t+1)} \le 1 + \log(T+1).
\end{cases}
$$

Here, we use the fact that $\int_t^{t+1} r(s)ds \le r(t) \le \int_{t-1}^{t} r(s)ds$ for a nonnegative and monotonically decreasing function $r$.

Substituting these estimates and $\sqrt{1-\beta_0} = \frac{1}{2^{1/3}}$ into (66), we eventually obtain

$$
\frac{1}{\sum_{t=0}^{T}\theta_t}\sum_{t=0}^{T}\theta_t\mathbb{E}\big[\|\mathcal{G}_{\eta_t}(x_t)\|^2\big] \le \frac{32P}{3\sqrt{b}\big[(T+3)^{2/3} - 2^{2/3}\big]}\big[\Psi_0(x_0) - \Psi_0^\star\big]
$$
$$
+ \frac{16Q}{3\big[(T+3)^{2/3} - 2^{2/3}\big]}\left[\frac{2^{1/3}}{\hat{b}_0} + \frac{2(1+\log(T+1))}{b}\right].
$$

Combining this inequality and $\frac{1}{\sum_{t=0}^{T}\theta_t}\sum_{t=0}^{T}\theta_t\mathbb{E}\big[\|\mathcal{G}_{\eta_t}(x_t)\|^2\big] = \mathbb{E}\big[\|\mathcal{G}_{\bar{\eta}_T}(\bar{x}_T)\|^2\big]$, we have proved (23) for $T \ge 0$. $\qquad\square$

## B.5   The proof of Theorem 3.3: The nonsmooth case with constant step-size

***The proof of Theorem 3.3 in the main text.*** Since $\mu_\psi = 0$, let us fix the smoothness parameter $\gamma_t = \gamma > 0$ and the weights $\beta_t = \hat{\beta}_t = \beta \in (0,1]$ for all $t \ge 0$. By Lemma A.1, we have

$$
M_{\phi_\gamma} = M_\psi\|K\|, \quad L_{\phi_\gamma} = \frac{\|K\|^2}{\gamma}, \quad \text{and} \quad L_{\Phi_\gamma} = L_F M_\psi\|K\| + \frac{M_F^2\|K\|^2}{\gamma}.
$$

Given batch sizes $b > 0$ and $\hat{b}_0 > 0$, for some $c_0 > 0$, let us also choose the mini-batch sizes as

$$
\hat{b}_1 = \hat{b}_2 := b, \quad b_1 = b_2 := \frac{c_0 b}{\gamma^2}, \quad \text{and} \quad b_0 := \frac{c_0\hat{b}_0}{\gamma^2}.
$$

Recall that $P$, $Q$, and $L_{\Phi_\gamma}$ are defined by (19). In this case, the quantities in (54) become

$$
\begin{cases}
\Theta_t := \Theta = \dfrac{M_F^2 L_{\phi_\gamma}\sqrt{26 b_1\hat{b}_1}}{3\big(\kappa M_F^4 L_{\phi_\gamma}^2\hat{b}_1 + \hat{\kappa}M_{\phi_\gamma}^2 L_F^2 b_1\big)^{1/2}} = \dfrac{\sqrt{26c_0 b}M_F^2\|K\|^2}{3\gamma(\kappa M_F^4\|K\|^4 + c_0\hat{\kappa}\|K\|^2 M_\psi^2 L_F^2)^{1/2}} & \overset{(19)}{=} \ \dfrac{M_F^2\|K\|^2 b^{1/2}}{\gamma P}, \\[4mm]
\Gamma_t := \Gamma = \dfrac{\sqrt{26 b_1\hat{b}_1}}{3\big(\hat{b}_1\kappa M_F^4 L_{\phi_\gamma}^2 + b_1\hat{\kappa}L_F^2 M_{\phi_\gamma}^2\big)^{1/2}}\left(\dfrac{\kappa M_F^2 L_{\phi_\gamma}^2\sigma_F^2}{b_2} + \dfrac{\hat{\kappa}M_{\phi_\gamma}^2\sigma_J^2}{b_2}\right) & \overset{(19)}{=} \ \dfrac{Q}{P\sqrt{b}}, \\[4mm]
\Pi_0 := \dfrac{\sqrt{26 b_1\hat{b}_1}}{3\big(\hat{b}_1\kappa M_F^4 L_{\phi_\gamma}^2 + b_1\hat{\kappa}L_F^2 M_{\phi_\gamma}^2\big)^{1/2}}\left(\dfrac{\kappa M_F^2 L_{\phi_\gamma}^2\sigma_F^2}{b_0} + \dfrac{\hat{\kappa}M_{\phi_\gamma}^2\sigma_J^2}{\hat{b}_0}\right) & \overset{(19)}{=} \ \dfrac{Q\sqrt{b}}{P\hat{b}_0}.
\end{cases}
$$

Furthermore, the step-sizes in (55) also become

$$\begin{cases} \theta_t & := \quad \theta = \frac{3L_{\Phi_\gamma}[b_1\hat{b}_1(1-\beta)]^{1/2}}{\sqrt{26}(\kappa M_F^4 L_{\phi_\gamma}^2 \hat{b}_1 + \hat{\kappa} M_{\phi_\gamma}^2 L_F^2 b_1)^{1/2}} \overset{(19)}{=} \frac{L_{\Phi_\gamma}[b(1-\beta)]^{1/2}}{P}, \\ \eta_t & := \quad \eta = \frac{2}{L_{\Phi_\gamma}(3+\theta)}. \end{cases}$$

Therefore, we have $\omega_t := \frac{\theta}{L_{\Phi_\gamma}}$ and

$$\Sigma_T := \sum_{t=0}^{T} \omega_t = \frac{\theta(T+1)}{L_{\Phi_\gamma}} = \frac{(T+1)[b(1-\beta)]^{1/2}}{P}.$$

Substituting these expressions into (57), we can further derive

$$\begin{aligned} \mathbb{E}\big[\|\mathcal{G}_\eta(\bar{x}_T)\|^2\big] &\leq \quad \frac{16P}{(T+1)[b(1-\beta)]^{1/2}} \Big(\mathbb{E}\big[\Psi_0(x_0) - \Psi_0^\star\big] + \gamma B_\psi\Big) \\ &\quad + 8Q\left[\frac{1}{\hat{b}_0(1-\beta)(T+1)} + \frac{2(1-\beta)}{b}\right]. \end{aligned} \tag{67}$$

From the last term of (67), we can choose $\beta$ as $\beta = 1 - \frac{b^{1/2}}{[\hat{b}_0(T+1)]^{1/2}}$. In this case, (67) reduces to

$$\mathbb{E}\big[\|\mathcal{G}_\eta(\bar{x}_T)\|^2\big] \leq \frac{16P\hat{b}_0^{1/4}}{[b(T+1)]^{3/4}}\Big(\mathbb{E}\big[\Psi_0(x_0) - \Psi_0^\star\big] + \gamma B_\psi\Big) + \frac{24Q}{[b\hat{b}_0(T+1)]^{1/2}}. \tag{68}$$

Clearly, from (68), to achieve the best convergence rate, we need to choose $\hat{b}_0 := c_1^2[b(T+1)]^{1/3}$. Then, since we choose $0 < \gamma \leq 1$ and $\mathbb{E}\big[\Psi_0(x_0)\big] = \Psi_0(x_0)$, (68) can be overestimated as

$$\mathbb{E}\big[\|\mathcal{G}_\eta(\bar{x}_T)\|^2\big] \leq \frac{\hat{\Delta}_0}{[b(T+1)]^{2/3}},$$

which proves (25), where $\hat{\Delta}_0$ is defined by (25), i.e.:

$$\hat{\Delta}_0 := 16P\sqrt{c_1}\big(\Psi_0(x_0) - \Psi_0^\star + B_\psi\big) + \frac{24Q}{c_1}.$$

Now, for any tolerance $\varepsilon > 0$, to obtain $\mathbb{E}\big[\|\mathcal{G}_\eta(\bar{x}_T)\|^2\big] \leq \varepsilon^2$, we require at most $T := \left\lfloor \frac{\hat{\Delta}_0^{3/2}}{b\varepsilon^3} \right\rfloor$ iterations. In this case, the total number of function evaluations $\mathcal{T}_F$ is at most

$$\mathcal{T}_F := b_0 + (T+1)(2b_1 + b_2) = \frac{c_0}{\gamma^2}c_1^2[b(T+1)]^{1/3} + \frac{3c_0}{\gamma^2}[b(T+1)] = \frac{c_0 c_1^2 \hat{\Delta}_0^{1/2}}{\gamma^2\varepsilon} + \frac{3c_0\hat{\Delta}_0^{3/2}}{\gamma^2\varepsilon^3}.$$

Alternatively, the total number of Jacobian evaluations $\mathcal{T}_J$ is at most

$$\mathcal{T}_J := \hat{b}_0 + (T+1)(2\hat{b}_1 + \hat{b}_2) = c_1[b(T+1)]^{1/3} + 3b(T+1) = \frac{c_1^2\hat{\Delta}_0^{1/2}}{\varepsilon} + \frac{3\hat{\Delta}_0^{3/2}}{\varepsilon^3}.$$

If we choose $\gamma := c_2\varepsilon$ for some $c_2 > 0$, then

$$\mathcal{T}_F := \frac{c_0 c_1^2 \hat{\Delta}_0^{1/2}}{c_2^2\varepsilon^3} + \frac{3c_0\hat{\Delta}_0^{3/2}}{c_2^2\varepsilon^5} = \mathcal{O}\left(\frac{\hat{\Delta}_0^{3/2}}{\varepsilon^5}\right),$$

which proves the last statement. $\square$

## B.6 The proof of Theorem 3.4: The nonsmooth case with diminishing step-size

***The proof of Theorem 3.4 in the main text.*** Using the fact that $\mu_\psi = 0$, from Lemma A.1, we have

$$M_{\phi_{\gamma_t}} = M_\psi\|K\|, \quad L_{\phi_{\gamma_t}} = \frac{\|K\|^2}{\gamma_t}, \quad \text{and} \quad L_{\Phi_{\gamma_t}} = L_F M_\psi\|K\| + \frac{M_F^2\|K\|^2}{\gamma_t},$$

where $\gamma_t > 0$, which will be appropriately updated. Moreover, let us choose $b_0 := \frac{c_0\hat{b}_0}{\gamma_0^2}$, $\hat{b}_1 = \hat{b}_2 := b$, and $b_1^t = b_2^t := \frac{c_0 b}{\gamma_t^2} > 0$, for some $b > 0$ and $c_0 > 0$. We also recall $P$, $Q$, and $L_{\Phi_\gamma}$ from (19).

With these expressions, the quantities defined by (54) and (55) become

$$
\begin{cases}
\theta_t &:= \dfrac{3L_{\Phi_{\gamma_t}}[b_1^t \hat{b}_1(1-\beta_t)]^{1/2}}{\sqrt{26}(\kappa M_F^4 L_{\phi_{\gamma_t}}^2 \hat{b}_1 + \hat{\kappa} M_{\phi_{\gamma_t}}^2 L_F^2 b_1^t)^{1/2}} \quad \overset{(19)}{=} \dfrac{L_{\Phi_{\gamma_t}}[b(1-\beta_t)]^{1/2}}{P}, \\[4mm]
\Theta_t &:= \dfrac{M_F^2 L_{\phi_{\gamma_t}}\sqrt{26 b_1^t \hat{b}_1}}{3\left(\kappa M_F^4 L_{\phi_{\gamma_t}}^2 \hat{b}_1 + \hat{\kappa} M_{\phi_{\gamma_t}}^2 L_F^2 b_1^t\right)^{1/2}} \quad \overset{(19)}{=} \dfrac{M_F^2 \|K\|^2 b^{1/2}}{\gamma_t P}, \\[4mm]
\Gamma_t &:= \dfrac{\sqrt{26 b_1^t \hat{b}_1}}{3\left(\hat{b}_1 \kappa M_F^4 L_{\phi_{\gamma_t}}^2 + b_1^t \hat{\kappa} L_F^2 M_{\phi_{\gamma_t}}^2\right)^{1/2}}\left(\dfrac{\kappa M_F^2 L_{\phi_{\gamma_t}}^2 \sigma_F^2}{b_2^t} + \dfrac{\hat{\kappa} M_{\phi_{\gamma_t}}^2 \sigma_J^2}{\hat{b}_2}\right) \quad \overset{(19)}{=} \dfrac{Q}{P\sqrt{b}}, \\[4mm]
\Pi_0 &:= \dfrac{\sqrt{26 b_1^0 \hat{b}_1}}{3\left(\hat{b}_1 \kappa M_F^4 L_{\phi_{\gamma_0}}^2 + b_1^0 \hat{\kappa} L_F^2 M_{\phi_{\gamma_0}}^2\right)^{1/2}}\left(\dfrac{\kappa M_F^2 L_{\phi_{\gamma_0}}^2 \sigma_F^2}{b_0} + \dfrac{\hat{\kappa} M_{\phi_{\gamma_0}}^2 \sigma_J^2}{\hat{b}_0}\right) \quad \overset{(19)}{=} \dfrac{Q\sqrt{b}}{P\hat{b}_0}.
\end{cases}
$$

Let us choose $\beta_t := 1 - \frac{1}{(t+2)^{2/3}} \in (0,1)$ and $\gamma_t := \frac{1}{(t+2)^{1/3}}$ as in (26). Then, it is easy to check that

$$
\frac{\beta_t^2(1-\beta_t)}{\Theta_t^2} \le \frac{1-\beta_{t+1}}{\Theta_{t+1}^2} \le \frac{1-\beta_t}{\Theta_t^2}.
$$

In addition, as before, one can show that

$$
\begin{cases}
\sum_{t=0}^T \sqrt{1-\beta_t} = \sum_{t=0}^T \frac{1}{(t+2)^{1/3}} \ge \int_2^{T+3} \frac{ds}{s^{1/3}} = \frac{3}{2}[(T+3)^{2/3} - 2^{2/3}], \\[3mm]
\sum_{t=0}^T \frac{(1-\beta_t)^2}{\sqrt{1-\beta_{t+1}}} = \sum_{t=0}^T \frac{(t+3)^{1/3}}{(t+2)^{4/3}} \le \sum_{t=0}^T \frac{1}{(t+1)} \le 1 + \log(T+1).
\end{cases}
$$

Using these estimates, we can easily prove

$$
\begin{cases}
\Sigma_T := \sum_{t=0}^T \omega_t &= \frac{\sqrt{b}}{P}\sum_{t=0}^T \sqrt{1-\beta_t} \ge \frac{3\sqrt{b}[(T+3)^{2/3}-2^{2/3}]}{2P}, \\[3mm]
\sum_{t=0}^T \frac{\Gamma_{t+1}(1-\beta_t)^2}{\sqrt{1-\beta_{t+1}}} &\le \frac{Q[1+\log(T+1)]}{P\sqrt{b}}
\end{cases}
$$

Substituting these inequalities into (57) and using $\sqrt{1-\beta_0} = \frac{1}{2^{1/3}}$, we further upper bound

$$
\begin{aligned}
\mathbb{E}\big[\|\mathcal{G}_\eta(\bar{x}_T)\|^2\big] \le\ & \frac{32P}{3\sqrt{b}[(T+3)^{2/3}-2^{2/3}]}\left(\Psi_0(x_0) - \Psi_0^\star + \frac{B_\psi}{(T+2)^{1/3}}\right) \\
& + \frac{16Q}{3[(T+3)^{2/3}-2^{2/3}]}\left(\frac{2^{1/3}}{\hat{b}_0} + \frac{2(1+\log(T+1))}{b}\right),
\end{aligned}
$$

which proves (27). $\qquad\square$

## C  Restarting variant of Algorithm 1 and its convergence and complexity

In this Supp. Doc., we propose a simple restarting variant, Algorithm 2, of Algorithm 1, prove its convergence, and estimate its oracle complexity bounds for both smooth $\phi_0$ and nonsmooth $\phi_0$ in (2). For simplicity of our analysis, we only consider the constant step-size case, and omit the diminishing step-size analysis.

### C.1  Restarting variant

**Motivation:** Since the constant step-size $\theta$ in (20) of Theorem 3.1 and (24) of Theorem 3.3 depends on the number of iterations $T$. Clearly, if $T$ is large, then $\theta$ is small. To avoid using small step-size $\theta$, we can restart Algorithm 1 by frequently resetting its initial point and parameters after $T$ iterations. This variant is described in Algorithm 2. Algorithm 2 has two loops, where each iteration $s$ of the outer loop is called the $s$-th stage. Unlike the outer loop in other variance-reduced methods relying on SVRG or SARAH estimators from the literature, which is mandatory to guarantee convergence, our outer loop is optional, since without it, Algorithm 2 reduces to Algorithm 1, and it still converges.

**Algorithm 2** (Restarting Variant of Algorithm 1)

---

1: **Inputs:** An arbitrarily initial point $\tilde{x}^0 \in \text{dom}(F)$, and a fixed number of iterations $T$.

2: **For** $s := 1, \cdots, S$ **do**

3:     Run Algorithm 1 for $T$ iterations starting from $x_0^{(s)} := \tilde{x}^{s-1}$.

4:     Set $\tilde{x}^s := x_{T+1}^{(s)}$ as the last iterate of Algorithm 1.

5: **EndFor**

6: **Output:** Choose $\bar{x}_N$ randomly from $\{x_t^{(s)}\}_{t=0 \to T}^{s=1 \to S}$ such that $\textbf{Prob}\left(\bar{x}_N = x_t^{(s)}\right) = \frac{\theta_t}{S\sum_{j=0}^{T}\theta_j}$.

---

## C.2 The smooth case $\phi_0$ with constant step-size

The smoothness of $\phi_0$ is equivalent to the $\mu_\psi$-strong convexity of $\psi$ in (1). The following theorem states convergence rate and estimates oracle complexity of Algorithm 2.

**Theorem C.1.** *Suppose that Assumptions 2.1 and 2.2 hold, $\psi$ is strongly convex (i.e., $\mu_\psi = 1 > 0$), and $P$, $Q$, and $L_{\Phi_0}$ are defined by (19). Let $\{x_t^{(s)}\}_{t=0 \to T}^{s=1 \to S}$ be generated by Algorithm 2 using $\gamma := 0$, $b_0 := c_0\hat{b}_0$, $b_1 = b_2 := c_0 b$, $\hat{b}_1 = \hat{b}_2 = b$ for some $c_0 > 0$ and given batch sizes $b > 0$ and $\hat{b}_0 > 0$, and the parameter configuration (20). Then, the following estimate holds*

$$\mathbb{E}\big[\|\mathcal{G}_\eta(\bar{x}_N)\|^2\big] \leq \frac{16P\hat{b}_0^{1/4}}{S[b(T+1)]^{3/4}}\big[\Psi_0(\tilde{x}^0) - \Psi_0^\star\big] + \frac{24Q}{[\hat{b}_0 b(T+1)]^{1/2}}, \tag{69}$$

*where $\bar{x}_N$ is uniformly randomly chosen from $\{x_t^{(s)}\}_{t=0 \to T}^{s=1 \to S}$.*

*Given $\varepsilon > 0$, if we choose $T := \left\lfloor\frac{48Q}{b\varepsilon^2}\right\rfloor$ and $\hat{b}_0 := \left\lfloor\frac{48Q}{\varepsilon^2}\right\rfloor$, then after at most $S := \left\lfloor\frac{8P}{\varepsilon\sqrt{3Q}}\right\rfloor$ outer iterations, we obtain $\mathbb{E}\big[\|\mathcal{G}_\eta(\bar{x}_N)\|^2\big] \leq \varepsilon^2$. Consequently, the total number of function evaluations $\mathcal{T}_F$ and the total number of Jacobian evaluations $\mathcal{T}_J$ are at most $\mathcal{T}_F = \mathcal{T}_J := \left\lfloor\frac{400P\sqrt{3Q}}{\varepsilon^3}\right\rfloor$.*

Theorem C.1 holds for any mini-batch $b$ such that $1 \leq b \leq \frac{48Q}{\varepsilon^2}$, which is different from, e.g., [43], where the complexity result holds under large batches. Moreover, the total oracle calls $\mathcal{T}_F$ and $\mathcal{T}_J$ are independent of $b$. In this case, the weight $\beta$ and the step-size $\theta$ become

$$\beta := 1 - \frac{b\varepsilon^2}{48Q} \quad \text{and} \quad \theta := \frac{bL_{\Phi_0}}{4P\varepsilon\sqrt{3Q}}.$$

Clearly, if $b$ is large, then our step-size $\theta$ is also large.

***The proof of Theorem C.1: Restarting variant.*** Since $\gamma := 0$, $\hat{b}_1 = \hat{b}_2 := b$ and $b_1 = b_2 := c_0 b$, from (61), using the superscript "$(s)$" for the outer iteration $s$, and $P$ and $Q$ from (19), we have

$$\frac{\theta}{16L_{\Phi_0}}\mathbb{E}\big[\|\mathcal{G}_\eta(x_t^{(s)})\|^2\big] \leq V_0(x_t^{(s)}) - V_0(x_{t+1}^{(s)}) + \frac{Q(1-\beta)^{3/2}}{Pb^{1/2}},$$

Summing up this inequality from $t := 0$ to $t := T$, and using the fact that $\tilde{x}^{s-1} := x_0^{(s)}$ and $\tilde{x}^s := x_{T+1}^{(s)}$, we get

$$\frac{\theta}{16L_{\Phi_0}}\sum_{t=0}^{T}\mathbb{E}\big[\|\mathcal{G}_\eta(x_t^{(s)})\|^2\big] \leq V_0(\tilde{x}^{s-1}) - V_0(\tilde{x}^s) + \frac{Q(T+1)(1-\beta)^{3/2}}{Pb^{1/2}}.$$

Using the choice $b_0 := c_0\hat{b}_0$, similar to the proof of (62), we can show that

$$\begin{aligned}
V_0(\tilde{x}^{s-1}) &= \mathbb{E}\big[\Psi_0(\tilde{x}^{s-1})\big] + \tfrac{\alpha}{2}\mathbb{E}\big[\|\tilde{F}_0^{(s)} - F(\tilde{x}^{s-1})\|^2\big] + \tfrac{\hat{\alpha}}{2}\mathbb{E}\big[\|\tilde{J}_0^{(s)} - F'(\tilde{x}^{s-1})\|^2\big] \\
&\leq \mathbb{E}\big[\Psi_0(\tilde{x}^{s-1})\big] + \frac{Qb^{1/2}}{2P\hat{b}_0\sqrt{1-\beta}}.
\end{aligned}$$

Using this estimate and and $V_0(\tilde{x}^s) \geq \Psi_0(\tilde{x}^s)$ into above inequality, we can further derive

$$\tfrac{1}{(T+1)} \sum_{t=0}^{T} \mathbb{E}\big[\|\mathcal{G}_\eta(x_t^{(s)})\|^2\big] \leq \tfrac{16 L_{\Phi_0}}{\theta(T+1)} \big[\Psi_0(\tilde{x}^{s-1}) - \Psi_0(\tilde{x}^s)\big] + \tfrac{16 Q L_{\Phi_0}(1-\beta)^{3/2}}{P\theta b^{1/2}}$$
$$+ \tfrac{8 Q L_{\Phi_0} b^{1/2}}{P\theta(T+1)\hat{b}_0 \sqrt{1-\beta}}.$$

Due to the choice of $b_1$ and $\hat{b}_1$, it follows from (20) that $\beta := 1 - \tfrac{b^{1/2}}{[\hat{b}_0(T+1)]^{1/2}}$ and $\theta := \tfrac{L_{\Phi_0} b^{3/4}}{P[\hat{b}_0(T+1)]^{1/4}}$. Therefore, the last inequality becomes

$$\frac{1}{(T+1)} \sum_{t=0}^{T} \mathbb{E}\big[\|\mathcal{G}_\eta(x_t^{(s)})\|^2\big] \leq \frac{16 P \hat{b}_0^{1/4}}{[b(T+1)]^{3/4}} \big[\Psi_0(\tilde{x}^{s-1}) - \Psi_0(\tilde{x}^s)\big] + \frac{24Q}{[\hat{b}_0 b(T+1)]^{1/2}}.$$

Summing up this inequality from $s := 1$ to $s := S$ and multiplying the result by $\tfrac{1}{S}$, we get

$$\frac{1}{S(T+1)} \sum_{s=1}^{S} \sum_{t=0}^{T} \mathbb{E}\big[\|\mathcal{G}_\eta(x_t^{(s)})\|^2\big] \leq \frac{16 P \hat{b}_0^{1/4}}{S[b(T+1)]^{3/4}} \big[\Psi_0(\tilde{x}^0) - \Psi_0(\tilde{x}^S)\big] + \frac{24Q}{[\hat{b}_0 b(T+1)]^{1/2}}.$$

Substituting $\Psi_0(\tilde{x}^S) \geq \Psi_0^\star$ into the last inequality, and using the fact that $\mathbb{E}\big[\|\mathcal{G}_\eta(\bar{x}_N)\|^2\big] = \tfrac{1}{S(T+1)} \sum_{s=1}^{S} \sum_{t=0}^{T} \mathbb{E}\big[\|\mathcal{G}_\eta(x_t^{(s)})\|^2\big]$, we obtain

$$\mathbb{E}\big[\|\mathcal{G}_\eta(\bar{x}_N)\|^2\big] = \tfrac{1}{S(T+1)} \sum_{s=1}^{S} \sum_{t=0}^{T} \mathbb{E}\big[\|\mathcal{G}_\eta(x_t^{(s)})\|^2\big]$$
$$\leq \tfrac{16 P \hat{b}_0^{1/4}}{S[b(T+1)]^{3/4}} \big[\Psi_0(\tilde{x}^0) - \Psi_0^\star\big] + \tfrac{24Q}{[\hat{b}_0 b(T+1)]^{1/2}},$$

which is exactly (69).

Now, for a given tolerance $\varepsilon > 0$, to obtain $\mathbb{E}\big[\|\mathcal{G}_\eta(\bar{x}_K)\|^2\big] \leq \varepsilon^2$, we need to impose

$$\frac{16 P \hat{b}_0^{1/4}}{S[b(T+1)]^{3/4}} = \frac{\varepsilon^2}{2} \quad \text{and} \quad \frac{24Q}{[\hat{b}_0 b(T+1)]^{1/2}} = \frac{\varepsilon^2}{2}.$$

This condition leads to $N = S(T+1) = \tfrac{32 P[\hat{b}_0(T+1)]^{1/4}}{b^{3/4}\varepsilon^2}$ and $\hat{b}_0 b(T+1) = \tfrac{48^2 Q^2}{\varepsilon^4}$. Hence, the total number of iterations is $N := S(T+1) = \tfrac{32 P[\hat{b}_0 b(T+1)]^{1/4}}{b\varepsilon^2} = \tfrac{128 P \sqrt{3Q}}{b\varepsilon^3}$.

Clearly, to optimize the oracle complexity, we need to choose $T+1 := \tfrac{48Q}{b\varepsilon^2}$, then $\hat{b}_0 := \tfrac{48Q}{\varepsilon^2}$ and $S := \tfrac{8P}{\sqrt{3Q}\varepsilon}$. In this case, the total number of function evaluations is at most

$$\mathcal{T}_F := b_0 S + 3bS(T+1) = \frac{48Q}{\varepsilon^2} \cdot \frac{8P}{\sqrt{3Q}\varepsilon} + 3bN = \frac{16 P \sqrt{3Q}}{\varepsilon^3} + \frac{384 P \sqrt{3Q}}{\varepsilon^3} = \frac{400 P \sqrt{3Q}}{\varepsilon^3}.$$

This is also the total number of Jacobian evaluations $\mathcal{T}_J$. $\qquad\square$

### C.3 The nonsmooth $\phi_0$ with constant step-size

Finally, we prove the convergence of Algorithm 2 when $\psi$ is non-strongly convex (i.e., $\phi_0$ in (2) is possibly nonsmooth).

**Theorem C.2.** *Assume that Assumptions 2.1 and 2.2 hold, $\psi$ in (1) is non-strongly convex (i.e., $\mu_\psi = 0$), and $P$, $Q$, and $L_{\Phi_\gamma}$ are defined by (19). Let $\{x_t^{(s)}\}_{t=0\to T}^{s=1\to S}$ be generated by Algorithm 2 after $N := S(T+1)$ iterations using:*

$$\begin{cases} b_1 = b_2 := \tfrac{2c_0 b \hat{R}_0}{\varepsilon^2}, & \hat{b}_1 = \hat{b}_2 := b, & b_0 := \tfrac{4c_0 \hat{R}_0^2}{\varepsilon^4}, & \hat{b}_0 := \tfrac{2\hat{R}_0}{\varepsilon^2}, \\ \gamma := \tfrac{\varepsilon}{\sqrt{2\hat{R}_0}}, & \text{and} \quad \beta := 1 - \tfrac{b\varepsilon^2}{2\hat{R}_0}. \end{cases} \tag{70}$$

*where $\varepsilon > 0$ is a given tolerance[1], and*

$$R_0 := 16\big[\Psi_0(\tilde{x}^0) - \Psi^\star + B_\psi\big] \quad \text{and} \quad \hat{R}_0 := 24Q. \tag{71}$$

Then, if we choose $T := \lfloor \frac{2\hat{R}_0}{\varepsilon^2} \rfloor$, then after at most $S := \lfloor \frac{\sqrt{2}R_0}{b\varepsilon\sqrt{\hat{R}_0}} \rfloor$ outer iterations, we obtain $\bar{x}_T$ such that $\mathbb{E}[\|\mathcal{G}_\eta(\bar{x}_T)\|^2] \le \varepsilon^2$.

Consequently, the total number of function evaluations $\mathcal{T}_F$ and the total number of Jacobian evaluations $\mathcal{T}_J$ are respectively at most

$$\mathcal{T}_F := \frac{4\sqrt{2}c_0 R_0 \hat{R}_0^{3/2}(3+b^{-1})}{\varepsilon^5} = \mathcal{O}\left(\frac{R_0\hat{R}_0^{3/2}}{\varepsilon^5}\right) \quad and \quad \mathcal{T}_J := \frac{2\sqrt{2}R_0\hat{R}_0^{1/2}(3+b^{-1})}{\varepsilon^3} = \mathcal{O}\left(\frac{R_0\hat{R}_0^{1/2}}{\varepsilon^3}\right).$$

**Remark C.1.** Note that we do not need to choose the batch sizes and parameters depending on $R_0$ as in (70), which is unknown since $\Psi_0^\star$ is unknown, but they are proportional to $R_0$. In this case, the complexity bounds in Theorem C.2 will only be shifted by a constant factor.

As we can see from Theorem C.2, the number of outer iterations $S$ is divided by the batch size $b$. However, the terms $\frac{12\sqrt{2}c_0 R_0 \hat{R}_0^{3/2}}{\varepsilon^5}$ and $\frac{6\sqrt{2}R_0\hat{R}_0^{1/2}}{\varepsilon^3}$ are independent of $b$ and dominate the complexity bounds in both $\mathcal{T}_F$ and $\mathcal{T}_J$, respectively.

***The proof of Theorem C.2.*** Let us first choose $\hat{b}_1 = \hat{b}_2 := b$, $b_1 = b_2 := \frac{c_0 b}{\gamma^2}$, and $b_0 := \frac{c_0 \hat{b}_0}{\gamma^2}$. With the same line as the proof of (67), we can show that

$$\frac{1}{(T+1)}\sum_{t=0}^T \mathbb{E}[\|\mathcal{G}_\eta(x_t^{(s)})\|^2] \le \frac{16P}{(T+1)[b(1-\beta)]^{1/2}}\left[\mathbb{E}[\Psi_0(x_0^{(s)})] - \mathbb{E}[\Psi_0(x_{T+1}^{(s)})] + \gamma B_\psi\right]$$
$$+ 8Q\left[\frac{1}{\hat{b}_0(1-\beta)(T+1)} + \frac{2(1-\beta)}{b}\right].$$

Here, we use the superscript "$(s)$" to present the outer iteration $s$. Moreover, instead of $\Psi_0^*$, we keep $\Psi_0(x_{T+1}^{(s)})$ from (63). Now, using the fact that $\tilde{x}^{s-1} = x_0^{(s)}$ and $\tilde{x}^s = x_{T+1}^{(s)}$, we can further derive from the above inequality that

$$\frac{1}{(T+1)}\sum_{t=0}^T \mathbb{E}[\|\mathcal{G}_\eta(x_t^{(s)})\|^2] \le \frac{16P}{(T+1)[b(1-\beta)]^{1/2}}\left[\mathbb{E}[\Psi_0(\tilde{x}^{s-1})] - \mathbb{E}[\Psi_0(\tilde{x}^s)] + \gamma B_\psi\right]$$
$$+ 8Q\left[\frac{1}{\hat{b}_0(1-\beta)(T+1)} + \frac{2(1-\beta)}{b}\right].$$

Summing up this inequality from $s := 1$ to $s := S$, and multiplying the result by $\frac{1}{S}$, and then using $0 < \gamma \le 1$, $\mathbb{E}[\Psi_0(\tilde{x}^0)] = \Psi_0(\tilde{x}^0)$, $\Psi_0(\tilde{x}^S) \ge \Psi_0^\star > -\infty$, and $\mathbb{E}[\|\mathcal{G}_\eta(\bar{x}_N)\|^2] = \frac{1}{S(T+1)}\sum_{s=1}^S\sum_{t=0}^T \mathbb{E}[\|\mathcal{G}_\eta(x_t^{(s)})\|^2]$, we arrive at

$$\mathbb{E}[\|\mathcal{G}_\eta(\bar{x}_N)\|^2] = \frac{1}{(T+1)S}\sum_{s=1}^S\sum_{t=0}^T \mathbb{E}[\|\mathcal{G}_\eta(x_t^{(s)})\|^2]$$
$$\le \frac{16P}{S(T+1)[b(1-\beta)]^{1/2}}\left[\Psi_0(\tilde{x}^0) - \Psi^\star + B_\psi\right]$$
$$+ 8Q\left[\frac{1}{\hat{b}_0(1-\beta)(T+1)} + \frac{2(1-\beta)}{b}\right].$$

Next, let us choose $\beta := 1 - \frac{b}{(T+1)}$ and $\hat{b}_0 := (T+1)$. Then, the above estimate becomes

$$\mathbb{E}[\|\mathcal{G}_\eta(\bar{x}_N)\|^2] \le \frac{16P}{bS(T+1)^{1/2}}\left[\Psi_0(\tilde{x}^0) - \Psi^\star + B_\psi\right] + \frac{24Q}{T+1}.$$

Let us define $R_0$ and $\hat{R}_0$ as in (71), i.e.:

$$R_0 := 16P[\Psi_0(\tilde{x}^0) - \Psi^\star + B_\psi] \quad and \quad \hat{R}_0 := 24Q.$$

In this case, for a given tolerance $\varepsilon > 0$, to achieve $\mathbb{E}[\|\mathcal{G}_\eta(\bar{x}_N)\|^2] \le \varepsilon^2$, we can impose

$$\frac{R_0}{bS(T+1)^{1/2}} = \frac{\varepsilon^2}{2} \quad and \quad \frac{\hat{R}_0}{(T+1)} = \frac{\varepsilon^2}{2}.$$

These conditions lead to $T + 1 = \frac{2\hat{R}_0}{\varepsilon^2}$ and $S := \frac{2R_0}{b(T+1)^{1/2}\varepsilon^2} = \frac{\sqrt{2}R_0}{b\varepsilon\sqrt{\hat{R}_0}}$. Let us also choose $\gamma := \frac{\varepsilon}{\sqrt{2\hat{R}_0}}$. Then, we also obtain the parameters as in (70), i.e.:

$$\begin{cases} b_1 = b_2 := \frac{2c_0 b\hat{R}_0}{\varepsilon^2}, & \hat{b}_1 = \hat{b}_2 := b, & b_0 := \frac{4c_0\hat{R}_0^2}{\varepsilon^4}, & \hat{b}_0 := \frac{2\hat{R}_0}{\varepsilon^2}, \\ \gamma := \frac{\varepsilon}{\sqrt{2\hat{R}_0}}, & and & \beta := 1 - \frac{b\varepsilon^2}{2\hat{R}_0}. \end{cases}$$

The total number $\mathcal{T}_F$ of function evaluations $\mathbf{F}(x_t^{(s)}, \xi_t)$ is at most

$$\mathcal{T}_F := S[b_0 + (T+1)(2b_1 + b_2)] = \frac{\sqrt{2}R_0}{b\varepsilon\sqrt{\hat{R}_0}}\left[\frac{4c_0\hat{R}_0^2}{\varepsilon^4} + \frac{2\hat{R}_0}{\varepsilon^2}\frac{6c_0 b\hat{R}_0}{\varepsilon^2}\right] = \frac{4\sqrt{2}c_0 R_0 \hat{R}_0^{3/2}}{\varepsilon^5}\left(\frac{1}{b} + 3\right).$$

The total number $\mathcal{T}_J$ of Jacobian evaluations $\mathbf{F}'(x_t^{(s)}, \xi_t)$ is at most

$$\mathcal{T}_J := S[\hat{b}_0 + (T+1)(2\hat{b}_1 + \hat{b}_2)] = \frac{\sqrt{2}R_0}{b\varepsilon\sqrt{\hat{R}_0}}\left[\frac{2\hat{R}_0}{\varepsilon^2} + \frac{6b\hat{R}_0}{\varepsilon^2}\right] = \frac{2\sqrt{2}R_0\hat{R}_0^{1/2}}{b\varepsilon^3} + \frac{6\sqrt{2}R_0\hat{R}_0^{1/2}}{\varepsilon^3}.$$

These prove the last statement of Theorem C.2. $\qquad\qquad\qquad\qquad\qquad\qquad\qquad\qquad\square$

## D  Experiment setup and additional experiments

This Supp. Doc. provides the details of configuration for our experiments in Section 4, and presents more numerical experiments to support our algorithms and theoretical results. As mentioned in the main text, all the algorithms used in this paper have been implemented in Python 3.6.3., running on a Linux desktop (3.6GHz Intel Core i7 and 16Gb memory).

Let us provide more details of our experiment configuration. We shorten the name of our algorithm, either Algorithm 1 or Algorithm 2, by Hybrid Stochastic Compositional Gradient, and abbreviate it by HSCG for both cases. We have implemented CIVR in [44] and ASC-PG in [38] to compare the smooth case of $\phi_0$. For the nonsmooth case of $\phi_0$, we have implemented two other algorithms, SCG in [37], and Prox-Linear in [34, 45]. While SCG only works for smooth $\phi_0$, we have smoothed it as in our method, and used the estimator as well as the algorithm in [37], but update the smoothness parameter as in our method. We also omit comparison in terms of time since Prox-Linear becomes slower if $p$ is large due to its expensive subproblem for evaluating the prox-linear operator. We only compare these algorithms in terms of epoch (i.e., the number of data passes).

Since both CIVR and ASC-PG are double loop, to be fair, we compare them with our restarting variant, Algorithm 2. To compare with SCG and Prox-Linear, we simply use Algorithm 1 since SCG has single loop. Since Prox-Linear requires to solve a nonsmooth convex subproblem, we have implemented a first-order primal-dual method in [5] to solve it. This algorithm has shown its efficiency in our test.

Note that the batch size $b$ is determined as $b := \lfloor\frac{N}{n_b}\rfloor$, where $N$ is the number of data points, and $n_b$ is the number of blocks. In our experiments, we have varied the number of blocks $n_b$ to observe the performance of these algorithms. Since we want to obtain the best performance, instead of using their theoretical step-sizes, we have carefully tuned the step-size $\eta$ of three algorithms in a given set of candidates $\{1, 0.5, 0.1, 0.05, 0.01, 0.001, 0.0001\}$. For our algorithms, we have another step-size $\theta_t$, which is also flexibly chosen from $\{0.1, 0.5, 1\}$. For the nonsmooth case, we update our smoothness parameter as $\gamma_t := \frac{1}{2(t+1)^{1/3}}$, which is proportional to the value in Theorems 3.2 and 3.4.

To further compare our algorithms with their competitors, we provide in the following subsections additional experiments for the two problems in the main text.

### D.1  Risk-averse portfolio optimization: Additional experiments

Figure 1 in the main text has shown the performance of three algorithms on three different datasets using 8 blocks, i.e., $n_b = 8$. Unfortunately, since ASC-PG does not work well when the number of blocks is larger than 8, we skip showing it in our comparison. To obverse more performance of HSCG and CIVR, we have increased the number of blocks $n_b$ from 8 to 32, 64, and 128. The convergence of the two algorithms is shown in Figure 3. As we can observe, HSCG remains slightly better than CIVR if $n_b = 32$ or 64. When $n_b = 128$, CIVR improves its performance and is slightly better than HSCG.

### D.2  Stochastic minimax problem: Additional experiments

For the stochastic minimax problem (32), Figure 2 has shown the progress of the objective values of three algorithms on three different datasets. Figure 4 simultaneously shows both the objective values and the gradient mapping norms of this experiment.

Figure 3: Comparison of two algorithms for solving (31) on larger blocks.

Figure 4: Comparison of three algorithms for solving (32) on 3 different datasets in Figure 2 with both objective values and gradient mapping norms.

Now, let us keep the same configuration as in Figure 2, but run one more case, where the number of blocks is increased to $n_b = 64$. The results are shown in Figure 5.

We again see that HSCG still highly outperforms the other two methods: SCG and Prox-Linear on **rcv1**. For **url**, HSCG is still slightly better than Prox-Linear as we have observed in Figure 2. However, for **covtype**, again, Prox-Linear shows a better performance than the other two competitors.

Note that since $p = 54$ in this dataset, we can solve the subproblem in `Prox-Linear` up to a high accuracy without incurring too much computational cost. Therefore, the inexactness of evaluating the prox-linear operator does not really affect the performance in this example.

Figure 5: Comparison of three algorithms for solving (32) on 64 blocks.

Finally, we test three algorithms: `HSCG`, `SCG`, and `Prox-Linear` on other three datasets: **w8a**, **phishing**, and **mushrooms** from LIBSVM [6]. We use the same number of blocks $n_b = 32$, and the results are reported in Figure 6. Figure 6 shows that `HSCG` highly outperforms both `SCG` and `Prox-Linear` on **w8a** and **phishing**. However, `Prox-Linear` becomes better than the other two on the **mushrooms** dataset.

Figure 6: Comparison of three algorithms for solving (32) on three more different datasets.

## Footnotes

[1]The batch sizes and $T$ in this paper must be integer, but for simplicity, we do not write their rounding form.