[Reviews · NeurIPS 2020]

Review 1

Summary and Contributions: This paper develops a variance-reduced algorithm to solve a stochastic nonconvex-concave minimax problem. From a computational point of view, this problem is challenging mainly due to its nonsmoothness and nonconvexity. The authors prove that the algorithm achieves the convergence of rate of O(T^{-2/3}) under standard assumptions and explain why their algorithms have several computational advantages compared to existing methods. Extensive experiments are conducted and the results confirm the benefits of the algorithms.

Strengths: 1. This paper is well motivated. Indeed, the stochastic optimization problem in Eq.(1) covers various application problems, including bimatrix games and adversarial learning. 2. Convergence analysis of Algorithm 1 is complete: both smooth and nonsmooth cases are discussed with constant/diminishing stepsizes. I briefly check the proof details in the appendix and find no mistake. 3. The authors did a great job in empirical evaluation. They compare their HSCG algorithm with four baseline approaches on two different examples. Experimental results on real datasets support their claims.

Weaknesses: 1. The title is misleading and the authors might overclaim their contribution. Indeed, the stochastic problem in Eq.(1) is a special instance of nonconvex-concave minimax problems and equivalent to nonconvex compositional optimization problem in Eq.(2). Solving such problem is easier than the general case consider in [23, 34]; see also (Rafique, Arxiv 1810.02060) and (Thekumparampil, NeurIPS'19). In addition, the KKT points and approximate KKT points are also defined based on such special structure. 2. The literature review is not complete. The authors mainly focus on the algorithms for stochastic compositional optimization instead of stochastic nonconvex-concave minimax optimization. 3. The algorithm is not single-loop in general. To be more specific, Algorithm 1 needs to solve Eq.(9) at each loop. This is also a nonsmooth strongly convex problem in general and the solution does not have the closed form. To this end, what is the advantage of Algorithm 1 over prox-linear algorithms in nonsmooth case? 4. Given the current stochastic problem in Eq.(1), I believe that the prox-linear subproblem can be reformulated using the conjugate function and becomes the same as the subproblem in Algorithm 1. That is to say, we can simply improve prox-linear algorithms for solving stochastic problem in Eq.(1). This makes the motivation of Algorithm 1 unclear. 5. The proof techniques heavily depend on the biased hybrid estimators introduced in [29]. The current paper does not convince me that such extension is nontrivial and has sufficient technical novelty.

Correctness: I believe the algorithm and Theorem 3.1-3.4 are mostly correct. The empirical methodology is also correct.

Clarity: Yes, the paper is well written.

Relation to Prior Work: The authors have compared their algorithm to stochastic compositional optimization algorithms, especially prox-linear algorithms. However, the argument is not convincing enough. The comparsion with general stochastic minimax optimization algorithms is missing.

Reproducibility: Yes

Additional Feedback: The main contribution of this paper is a single-loop stochastic algorithm which achieves the best-known complexity bound and outperforms the existing prox-linear algorithms in computational sense. To make this argument more convincing, I hope the authors can address the following concerns: 1. Clearly state why your algorithm outperforms prox-linear algorithms. Indeed, by simply exploring the structure of stochastic problem in Eq.(1), the prox-linear subproblem can be reformulated using conjugate function and becomes the same as your subproblem. 2. Subproblem solving with Eq.(9). Indeed, when b and \psi are in special forms, I agree that we can achieve the closed-form solution as you point out. However, this also holds true for prox-linear algorithms and a few other double-loop algorithms. Thus, why is your algorithm single-loop in general? 3. Clearly explain why your convergence analysis is novel enough to merit the acceptance. Indeed, I agree that the biased hybrid estimators are introduced in [29] and indeed novel. However, the proofs in this paper seems a rather straightforward extension. To this end, I encourage the authors to provide a brief overview of your technical contribution during feedback. 4. Change the title to the stochastic compositional optimization or add the references and discussions on general nonconvex-concave minimax optimization problems. I suggest so since your paper mainly focuses on the algorithms for solving such problem which is a special and relatively easy instance of nonconvex-concave minimax optimization problems. If the authors still hope to keep the title with nonconvex-concave minimax problems, I encourage you to add some comments on KKT points, changing the writing style and cite the relevant references, e.g., (Rafique, Arxiv 1810.02060), (Lu et.al. Arxiv 1902.08294), (Thekumparampil, NeurIPS'19), (Kong and Monteiro, Arxiv 1905.13433), (Lin et.al. Arxiv 1906.00331), (Yan et.al. Arxiv 2002.05309), (Xu et.al. Arxiv 2006.02032) and so on. I am happy to raise my score if the authors can address my concerns. ============================================================== I have read the rebuttals. The authors convince me that the closed-form solution of prox-linear subproblems (both primal and dual forms) only exist in special cases and that their algorithm can be single-loop when b is chosen as the quadratic function and the proximal mappings of \psi and R both have the closed-form solution. Thus, I decide to raise my score from 5 to 6.


Review 2

Summary and Contributions: Paper proposes and analyzes a variance reduced algorithm for nonconvex-concave minimax problem of the form: min_x max_y R(x) + <Ky, E[F(x, \xi)]> - \psi(y) where the coupling is nonlinear x and linear in y and the expectation is E is taken over a random variable \xi. It is also assumed that proximal operators of R and \psi are easily computable. The algorithm uses (a) Nesterov’s smoothing technique and (b) Hybrid variance-reduced stochastic gradient estimators [29] to obtain a single-loop algorithm converging to an approximate KKT point.

Strengths: 1. Authors provide a single-loop variance-reduced algorithm which can handle variable batchsize. 2. The algorithm followed by a post-processing procedure computes an eps-KKT point using only (a) O(1/eps^3) minibatched F(x, \xi) and and its minibatched Jacobian-vector evaluations when \psi is strongly convex (b) O(1/eps^5) minibatched F(x, \xi) evaluations and O(1/eps^3) minibatched Jacobian-vector evaluations when \psi is not strongly convex 3. For the <Ky, E[F(x, \xi)]> part of the objective, earlier methods either (a) assumed that max_y <Ky, E[F(x, \xi)]> - \psi(y) and F(x) are smooth, or (b) only required smoothness of F(x) but assumed access to prox-linear operators. The new method only assumes F(x) is smooth and access to stochastic Jacobian of F(x). But it still achieves similar performance as previous methods. 4. Good discussion of potential broader impact.

Weaknesses: 1. From the presentation it not super clear which technical contributions are novel and which aspects of the algorithms and analysis are put together from previously known ideas. 2. There is not enough discussion and justification of the various choices made in the algorithm like: (a) why use hybrid stochastic gradient (b) two different stepsizes \etat_t and \theta_t (b) how the proposed method is able to use only a single loop as compared to say [38] 3. Assumption that proximal operators of R & \psi and Fenchel conjugate of \psi are easily computable (not a trivial statement) needs to be explicitly stated as an assumption. Similarly, assumption that the randomness in \xi (14) is not blackbox and is somewhat controllable (as in ERM sense) needs to be explicitly stated. 4. Discussion of related work on nonconvex-concave problems with nonlinear coupling is missing [a, b, c, d](see Q.relation) which seems to be more general than the setting in the paper. Some of these work does not seem to make more assumptions like PL condition, smoothness of phi_0, or access to prox-linear operator. Can the authors compare the their results to these? For example the deterministic version of Sec 4.2 seems very similar to the one analyzed in [b]. 5. How the proposed method relates to or differs from other methods is not super clear. For example, details of CIVR used in experiments are not presented. 6. Authors could motivate the problem formulation better. E.g. mention model selection in the beginning itself. —————— UPDATE: I thank the authors for their response. Authors reasonably addressed my queries regarding novelty, algorithmic choices, and implicit assumptions. I hope appropriate modifications would be made in the next revision.

Correctness: The theoretical claims seem to be correct under initial scrutiny. Empirical methodology could be improved

Clarity: Clarity of the paper could be improved (see other comments)

Relation to Prior Work: 1. From the presentation it not clear which technical contributions are novel and which aspects of the algorithms and analysis are put together from previously known ideas. 2. Discussion of related work on nonconvex-concave problems with nonlinear coupling is missing: [a] Kong, and Monteiro, An accelerated inexact proximal point method for solving nonconvex-concave min-max problems, arxiv 2019 [b] Thekumparampil, Jain, Netrapalli, and Oh, Efficient algorithms for smooth minimax optimization, NeurIPS 2019 [c] Lin, Jin, and Jordan, On gradient descent ascent for nonconvex-concave minimax problems, arxiv 2019 [d] Zhao, A Primal Dual Smoothing Framework for Max-Structured Nonconvex Optimization, arxiv 2020 Some of these work does not seem to make more assumptions like PL condition, smoothness of phi_0, or access to prox-linear operator. Can the authors compare the their results to these? 3. How the proposed method relates to or differs from other methods is not super clear. For example, details of CIVR used in experiments are not presented.

Reproducibility: Yes

Additional Feedback: See Q.weakness for major comments. 1. Sec 3.2 has too many (complicated) notations and equations making the theorems (and analysis) harder to interpret. It might be best to provide informal statement to convey the gist in the main text, move the formal statements to the appendix and use the extra space to provide better intuition for the algorithm and theory. Authors could also move variable stepsize algorithms to the appendix if more space is needed. 2. It would be better if the plots are in loglog or semilogx scale so that algorithms can be compared fairly. Best would be to plot suboptimality gap f(x_t) - f_min versus iteration t to be plotted in loglog scale. 3. L_{\Phi_{\gamma}} is used in Algorithm 1 before its definition. 4. Instead of objective fn vs number of iterates, a better comparison would be plotting objective vs number of equivalent one sample batched jacobian and function computed. Similarly objective vs time (seconds) could provide a good comparison. 5. Author might want to make it explicit that since \phi_0 is smooth and explicit we don’t need Nesterov smoothing in Sec 4.1.


Review 3

Summary and Contributions: This paper considers a class of composite nonconvex optimization problems, and obtains the best known oracle complexities in two cases, i.e., the outer function is smooth and nonsmooth.

Strengths: It seems that the main contribution is to obtain a best known oracle complexity when \psi is non-strongly convex. When \psi is strongly convex, the complexity is the same as the existing work.

Weaknesses: This paper has no serious weaknesses. See my comments below.

Correctness: Yes, although I didn't verify carefully.

Clarity: Moderate.

Relation to Prior Work: Yes, it is clear.

Reproducibility: Yes

Additional Feedback: 1. I am wondering how difficult it is to obtain the complexity results when \psi is non-strongly convex? It seems that a standard dual smoothing technique is used, which makes \psi strongly convex. 2. I think the authors shouldn't use the phrase nonconvex-concave in the title, because the coupling function is linear in y. The complexity results can be very different in the non-linear case. Although most real examples are linear in y, from a theoretical point of view, probably you should consider nonconvex-linear instead. 3. I don't think single loop, constant stepsize, and sinlge sample are advantages of the proposed method, at least not theoretically, unless you can show that these features result in superior oracle complexities.


Review 4

Summary and Contributions: This paper proposes a new variance-reduced algorithm to optimize a certain class of nonconvex-concave minimax problems, where the problem can be reformulated as a stochastic compositional non-convex problem. The authors provide a theoretical analysis of the convergence rate and empirically evaluate the performance of the algorithm comparing to existing works [31] [39].

Strengths: This paper proposes a variance-reduced algorithm to optimize a certain class of nonconvex-concave minimax problems mainly based on the variance reduction technique in [29]. Comparing to existing works, the proposed method has single loop, can work with either single sample or mini-batch and allows both constant or diminishing step-sizes.

Weaknesses: When $\psi$ is strongly convex, this problem considered in the paper becomes a nonconvex strongly concave minimax problem. Recently, there are several papers on this topic, for example, "Stochastic Recursive Gradient Descent Ascent for Stochastic Nonconvex-Strongly-Concave Minimax Problems". This paper also achieves an O(epsilon^-3 ) rate. The authors did not compare their results with this paper. The convergence results in theorems are clear in general. However, there are many hyperparameters defined, and some of them are set to be equal all the time. Is it possible to show them in a more concise way? In addition, the $c_0$ and $c_1$ seems not clearly defined in the main text, which makes the presentation of the theorems a little confusing. Do $c_0$ and $c_1$ depend on other hyperparameters in the theorems? In this paper, the authors claim that the proposed method can work with either a single sample or mini-batches. However, as shown in theorem 3.1, it seems that this method still needs a large batch for initialization. Is this large-batch initialization process necessary? Typos: (1) In Eq (9), $b$ denotes a function while in theorems (for example, Theorem 3.1) $b$ is a constant. (2) Line 124 "the expectation is taken overall" --> "the expectation is taken over all"

Correctness: The claims look correct. The detailed proofs are not carefully checked. The empirical methodology is correct.

Clarity: The writing of this paper is good in general. It can be improved if the authors can present the theorems in a more clear way.

Relation to Prior Work: The authors have clearly discussed the contributions different from previous works listed in this paper. The recent work "Stochastic Recursive Gradient Descent Ascent for Stochastic Nonconvex-Strongly-Concave Minimax Problems" needs to be further discussed.

Reproducibility: Yes

Additional Feedback:

[Author Response · NeurIPS 2020]

We thank all the reviewers and ACs for handling our paper and their constructive comments. Due to space limit, we will
focus on responding to the main concerns. Other minor points will be carefully addressed in our next version.
**R1:** 1) **(9) vs. prox-linear:** We are sorry if our explanation in the submission was unclear. We cannot choose $\psi$, which
comes from (1). But we have the freedom to choose $b$ in (9), which does not depend on problem (1) or (2). If we choose
$b(y) := \frac{1}{2}\|y - \dot{y}\|^2$ for any given $\dot{y}$, then (9) becomes $\text{prox}_{\psi/\gamma}(\cdot)$ as explained in line 132 of the submission. In contrast,
the prox-linear operator for (2) requires to solve $\min_x\{\phi_0(\tilde{F}_t + \tilde{J}_t(x - x_t)) + \mathcal{R}(x)\}$ (see [28,39]), which does not have
closed form solution in general even using proximal operators of $\phi_0$ and $\mathcal{R}$. This is due to the composition between $\phi_0$
and $\tilde{F}_t + \tilde{J}_t(x - x_t)$. If we use Fenchel conjugate, the dual problem is $\min_z\{\mathcal{R}^*(-\tilde{J}_t^T z) + \phi_0^*(z) - \langle \tilde{F}_t - \tilde{J}_t x_t, z\rangle\}$,
which still does not have closed form solution (see [28,39] for more discussion). Hence, evaluating the prox-linear
operator requires solving this complex subproblem (e.g., using primal-dual methods). As a result, the per-iteration
complexity of Alg. 1 is better than prox-linear-based methods. Numerical experiments also reveal such a difference.
2) **Why single loop?** As explained, we can choose the quadratic $b$ to have closed form $y^*_{\gamma_t}(\tilde{F}_t) = \text{prox}_{\psi/\gamma_t}(\dot{y} - $
$\gamma_t^{-1} K^T \tilde{F}_t)$. Hence, Alg. 1 is a single loop. Note that variance-reduced algorithms with prox-linear operators, e.g., in
[28,39], have double loops regardless of the computation of prox-linear operator. As explained, the prox-linear operator
often does not have a closed-form. If we solve it with an additional loop, then these algorithms even have three loops.
3) **Convergence analysis:** Since we exploit the hybrid estimators (14) from [29], Lemma A.2 is indeed adapted from
[29]. Lemma B.1 has a similar proof as in [29], but the relation is between $\Psi_{\gamma_t}$ and $\Psi_{\gamma_{t-1}}$, which requires new result
(e) of Lemma A.1. We believe that all other technical proofs are new and do not overlap or recycle from [29]. In fact,
Lemmas A.2. and B.1 are not our main results, but Th. B.1 is the key step to prove Th. 3.1 to Th. 3.4. We believe that
the proof of Th. B.1. is non-trivial and requires significant technical details and mathematical derivations. Moreover,
the adaptive weight $\beta_t$ is new, and it can remove the initial batches $b_0$ and $\hat{b}_0$ requirement in Alg. 1 though it sacrifices a
log factor in convergence rate. We will highly appreciate it if the reviewer could take some time to check our proofs.
4) **Title and literature review:** We will adapt the title to make it more precise. Due to the space limit, we did not have
much chance to add full literature review. We will add those references and discuss KKT points in our next version.
**R3: Weaknesses concerns:** 1) We believe that our paper has significant novelty compared to previous works. For
instance, treating nonstrongly convex $\psi$ with a variance-reduced, single-loop method is new. Analyzing complexity
with a single sample, a wide range of mini-batches $b$, adaptive $\beta_t$, and diminishing stepsize is also new.
2) 3) 6) We will certainly implement your suggestions. Due to space limit, we are unable to answer these in detail here.
We briefly respond to 2): (a) Hybrid estimator can trade-off variance and bias. Two step-sizes can handle the regularizer
$\mathcal{R}$. Obtaining single-loop is due to properties of hybrid estimator (see Lemma A.2) compared to others, like [38].
4) Thank you. Indeed, we were rather selective due to the space limit and certainly missed some references. We were
aware of [a], [b], and [d], but since they are deterministic though [d] has small stochastic part, we probably skip them.
We will add and discuss them in our next version. The algorithms in [a,b,d] are double or triple loops, more complicated
than Alg. 1. The stochastic algorithm in [c] is a single loop but has a worse oracle complexity than ours. These works
indeed do not need the PL condition, but they use stronger or different assumptions than ours (see, e.g., (1.2) in [d]).
5) We will elaborate on our comparison in the next version. CIVR uses SARAH (Nguyen et al., 2017) estimator, which
has two loops. The details of CIVR in the numerical experiments are given in Supp. Doc. D.
**Feedback:** 1) We will implement your suggestions and make the paper more readable. 2) Usually, researchers compare
stochastic methods via epochs, but we appreciate your suggestion and will add some figures on the loglog scale.
**R4:** 1) If $\psi$ in (1) is non-strongly convex, then $\phi_0$ is nonsmooth, and gradient-based methods are not applicable. Indeed,
the smoothing technique is used to make $\phi_0$ smooth, but it changes the original problem. This technique is still new
when solving complex models (1) or (2). Moreover, we can adaptively update the smoothness parameter $\gamma$ instead of
choosing a tiny value such as $\gamma = \mathcal{O}(\varepsilon)$ as often seen in smoothing techniques (see Th. 3.4).
2) We will adapt the title as suggested.
3) We believe the single loop, constant stepsize, and different batches $b$ are advantages, but we will carefully implement
your suggestion. We think that $\mathcal{O}(\varepsilon^{-3})$-complexity is optimal (see [2] for details) for the strongly convex $\psi$. For
nonstrongly convex $\psi$, our result seems to be the first so far without using prox-linear operator, and achieves the
best-known oracle complexity bound. Alg. 1. works with a wide range of batches $b$, not only one choice as [37,38].
**R5:** 1) We apologize for missing "Stochastic ... Problems" paper (which is almost concurrent to our work), we will cite
it. This paper is indeed similar to [37] but treats a general nonconvex-strongly concave minimax problem. Compared to
this paper, Alg. 1 has a single-loop instead of two loops as SREDA in that paper. Alg. 1 still works under a single
sample, different mini-batch sizes, and constant and diminishing stepsizes instead of specific mini-batch as in SREDA
to achieve such an $\mathcal{O}(\varepsilon^{-3})$ rate. Alg. 1 also tackles the nonstrongly convex $\psi$, which is new and perhaps harder.
2) We will try to facilitate some parameters as suggested. In theory, we only need $b$ and $\hat{b}_0$, while $c_0$ and $c_1$ can be fixed.
Other parameters are explicitly defined through $b$ and $\hat{b}_0$ (e.g., (19)). Here, $c_1$ and $c_0$ do not depend on any parameter.
3) Indeed, Th.3.1 requires a large initial mini-batch, but it is used only once as opposed to each iteration as in other
works, e.g., [37,38,39]. Th. 3.2 does not require a large initial mini-batch (see line 181).
4) Thank you for indicating typos. We will correct them.

[Meta-Review · NeurIPS 2020]

The paper introduces a single-loop stochastic algorithm for solving a special class of nonconvex-concave minimax problems that achieves best-known complexity bound. The rebuttal addressed most of the reviewers' concerns on the algorithmic justification, although some concern remains in terms of the special structure. I recommend acceptance. However, please consider revising the paper to address R1 and R3 's remarks, in particular: - Adjust the title to reflect the special structure instead of overclaim the contribution; - Elaborate the desirable property of single-loop algorithm over existing methods; - Add detailed comparisons to prior work including prox-linear algorithms for compositional problems and recent algorithms for general nonconvex-concave minimax problems.